# Unleashing the Power of Contrastive Self-Supervised Visual Models via Contrast-Regularized Fine-Tuning

**Yifan Zhang**[1][*]   **Bryan Hooi**[1]   **Dapeng Hu**[1]   **Jian Liang**[2]   **Jiashi Feng**[3]
[1]National University of Singapore   [2]Chinese Academy of Sciences   [3]SEA AI Lab

## Abstract

Contrastive self-supervised learning (CSL) has attracted increasing attention for model pre-training via unlabeled data. The resulted CSL models provide instance-discriminative visual features that are uniformly scattered in the feature space. During deployment, the common practice is to directly fine-tune CSL models with cross-entropy, which however may not be the best strategy in practice. Although cross-entropy tends to separate inter-class features, the resulting models still have limited capability for reducing intra-class feature scattering that exists in CSL models. In this paper, we investigate whether applying contrastive learning to fine-tuning would bring further benefits, and analytically find that optimizing the contrastive loss benefits both discriminative representation learning and model optimization during fine-tuning. Inspired by these findings, we propose Contrast-regularized tuning (Core-tuning), a new approach for fine-tuning CSL models. Instead of simply adding the contrastive loss to the objective of fine-tuning, Core-tuning further applies a novel hard pair mining strategy for more effective contrastive fine-tuning, as well as smoothing the decision boundary to better exploit the learned discriminative feature space. Extensive experiments on image classification and semantic segmentation verify the effectiveness of Core-tuning.

## 1   Introduction

Pre-training a deep neural network on a large database and then fine-tuning it on downstream tasks has been a popular training scheme. Recently, contrastive self-supervised learning (CSL) has attracted increasing attention on model pre-training, since it does not rely on any hand-crafted annotations but even achieves more promising performance than supervised pre-training on downstream tasks [6, 7, 20, 22, 47]. Specifically, CSL leverages unlabeled data to train visual models via contrastive learning, which maximizes the feature similarity for two augmentations of the same instance and minimizes the feature similarity of two instances [58]. The learned models provide instance-discriminative visual representations that are uniformly scattered in the feature space [53].

Although there have been substantial CSL studies on model pre-training [23, 47], few have explored the fine-tuning process. The common practice is to directly fine-tune CSL models with the cross-entropy loss [6, 13, 20]. However, we empirically (cf. Table 1) find that different fine-tuning methods significantly influence the model performance on downstream tasks, and fine-tuning with only cross-entropy is not the optimal strategy. Intuitively, although cross-entropy tends to learn separable features among classes, the resulting model is still limited in its capability for reducing intra-class feature scattering [37, 55] that exists in CSL models. Meanwhile, most existing fine-tuning methods [33, 35] are devised for supervised pre-trained models and tend to enforce regularizers to prevent the fine-tuned models changing too much from the pre-trained ones. However, they suffer from the issue of negative transfer [9], since downstream tasks are often different from the pre-training contrastive task. In this sense, how to fine-tune CSL models remains an important yet under-explored question.

---

[*]Corresponding to: Yifan Zhang <yifan.zhang@u.nus.edu>

35th Conference on Neural Information Processing Systems (NeurIPS 2021).

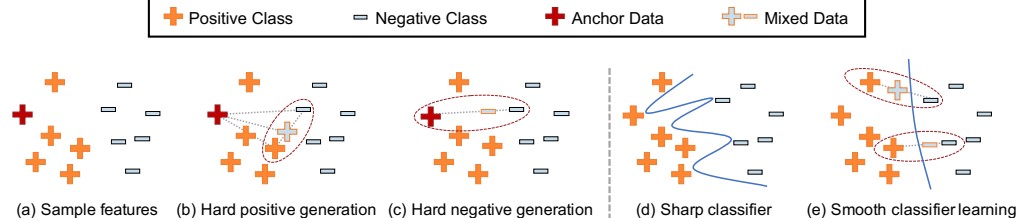

Figure 1: Illustration of two challenges in contrastive fine-tuning. (1) How to mine hard sample pairs for more effective contrastive fine-tuning. As shown in (a), the majority of sample pairs are easy-to-contrast, which may induce negligible contrastive loss gradients that contribute little to learning discriminative representations. (2) How to improve the generalizability of the model. As shown in (d), the classifier simply trained with cross-entropy is often sharp and near training data, leading to limited generalization performance.

Considering that optimizing the unsupervised contrastive loss during pre-training yields models with instance-level discriminative power, we investigate whether applying contrastive learning to fine-tuning would bring further benefits. To answer this, we analyze the contrastive loss during fine-tuning (cf. Section 3) and find that it offers two benefits. First, integrating the contrastive loss into cross-entropy can provide an additional regularization effect, as compared to cross-entropy based fine-tuning, for discriminative representation learning. Such an effect encourages the model to learn a low-entropy feature cluster for each class (*i.e.,* high intra-class compactness) and a high-entropy feature space (*i.e.,* large inter-class separation degree). Second, optimizing the contrastive loss will minimize the infimum of the cross-entropy loss over training data, which can provide an additional optimization effect for model fine-tuning. Based on the optimization effectiveness as well as the regularization effectiveness on representations, we argue that optimizing the contrastive loss during fine-tuning can further improve the performance of CSL models on downstream tasks.

Considering the above benefits, a natural idea is to directly add the contrastive loss to the objective for fine-tuning, *e.g.,* one recent study [18] simply uses contrastive learning to fine-tune language models. However, such a method cannot take full advantage of contrastive learning, since it ignores an important challenge in contrastive fine-tuning. That is, contrastive learning highly relies on positive/negative sample pairs, but the majority of sample features are easy-to-contrast (cf. Figure 1 (a)) [19, 56] and may produce negligible contrastive loss gradients. Ignoring this makes the method [18] fail to learn more discriminative features via contrastive learning and thus cannot fine-tune CSL models well.

In this paper, to better fine-tune CSL models and enhance their performance on downstream tasks, we propose a contrast-regularized tuning approach (termed Core-tuning), based on a novel hard pair mining strategy. Specifically, Core-tuning generates both hard positive and hard negative pairs for each anchor data via a new hardness-directed mixup strategy (cf. Figure 1 (b-c)). Here, hard positives indicate the positive pairs far away from the anchor, while hard negatives are the negative pairs close to the anchor. Meanwhile, since hard pairs are more informative for contrastive learning [19], Core-tuning further assigns higher importance weights to hard positive pairs based on a new focal contrastive loss. In this way, the resulting model is able to learn a more discriminative feature space by contrastive fine-tuning. Following that, we also explore how to better exploit the learned discriminative feature space in Core-tuning. Previous work has found that the decision boundary simply trained with cross-entropy is often sharp and close to training data [52], which may make the classifier fail to exploit the high inter-class separation degree in the discriminative feature space (cf. Figure 1 (d)), and also suffer from limited generalization performance. To address this, Core-tuning further uses the mixed features to train the classifier, so that the learned decision boundaries can be more smooth and far away from the original training data (cf. Figure 1 (e)).

The key contributions are threefold. 1) To our knowledge, we are among the first to look into the fine-tuning stage of CSL models, which is an important yet under-explored question. To address this, we propose a novel Core-tuning method. 2) We theoretically analyze the benefits of the supervised contrastive loss on representation learning and model optimization, revealing that it is beneficial to model fine-tuning. 3) Promising results on image classification and semantic segmentation verify the effectiveness of Core-tuning for improving the fine-tuning performance of CSL models. We also empirically find that Core-tuning benefits CSL models in terms of domain generalization and adversarial robustness on downstream tasks. Considering the theoretical guarantee and empirical effectiveness of Core-tuning, we recommend using it as a standard baseline to fine-tune CSL models.

## 2 Related Work

**Contrastive self-supervised learning (CSL).** Self-supervised learning is a kind of unsupervised learning method based on self-supervised proxy tasks, *e.g.,* rotation prediction [16], colorization prediction [30] and clustering [60]. Recently, CSL has become the most popular self-supervised paradigm, which treats each instance as a category to learn instance-discriminative representations. State-of-the-art CSL methods include InsDis [58], MoCo [20], SimCLR [6, 7] and InfoMin [47]. Most CSL studies are devoted to network pre-training, but few have explored the fine-tuning process.

As an effective data augmentation method, mixup [66] has recently been applied to instance augmentation for CSL [23, 26, 31, 45]. Among these methods, the work [23] uses mixup to generate hard negative pairs for better instance discrimination. However, all these methods focus on unsupervised pre-training and cannot accurately generate hard pairs regarding classes. Comparatively, Core-tuning focuses on the fine-tuning of CSL models and can generate accurate hard positive/negative pairs for each class. Note that the hardness-directed mixup strategy in Core-tuning is different from manifold mixup [52] that cannot be directly used to generate hard sample pairs.

**Pre-training and Fine-tuning.** In deep learning, it is a popular scheme to first pre-train a deep neural network on a large database (*e.g.,* ImageNet) and then fine-tune it on downstream tasks [35, 34]. Supervised learning is the mainstream method for pre-training [27], whereas self-supervised learning is attracting increasing attention since it does not rely on rich annotations [6, 7]. Most existing methods for fine-tuning, like L2-SP [35] and DELTA [33], are devised for supervised pre-trained models and tend to enforce some regularizer to prevent the fine-tuned models changing too much from the pre-trained ones. However, they may be unsuitable for contrastive self-supervised models, since downstream tasks are often different from the contrastive pre-training task, leading to negative transfer [9]. Very recently, one work [18] explored contrastive learning to fine-tune language models. However, it simply add the contrastive loss to the objective of fine-tuning and cannot theoretically explain why it boosts fine-tuning. More critically, it ignores the challenge of hard pair mining in contrastive fine-tuning and thus cannot fine-tune CSL models well.

## 3 Effects of Contrastive Loss for Model Fine-tuning

We start by analyzing the benefits of the contrastive loss during fine-tuning, which will motivate our new method. Before that, we first define the problem and notations.

**Problem Definition and Notation.** This paper studies the fine-tuning of contrastive self-supervised visual models that are pre-trained on a large-scale unlabeled database. During fine-tuning, let $\{(x_i, y_i)\}_{i=1}^n$ denote the target task dataset with $n$ samples, where $x_i$ is an instance with one-hot label $y_i \in \mathbb{R}^K$ and $K$ denotes the number of classes. The neural network model is denoted by $G$, which consists of a pre-trained feature encoder $G_e$ and a new predictor $G_y$ specific to the target task. Based on the network, we extract visual representations by $z_i = G_e(x_i)$ and make a prediction by $\hat{y}_i = G_y(z_i)$. Such a contrastive self-supervised model is generally fine-tuned with the cross-entropy loss [13, 20].

Following [1], we define the random variables of samples and labels as $X$ and $Y$, and those of embeddings and predictions as $Z|X \sim G_e(X)$ and $\hat{Y}|Z \sim G_y(Z)$, respectively. Moreover, let $p_Y$ be the distribution of $Y$, $p_{(Y,Z)}$ be the joint distribution of $Y$ and $Z$, and $p_{Y|Z}$ be the conditional distribution of $Y$ given $Z$. We define the entropy of $Y$ as $\mathcal{H}(Y) := \mathbb{E}_{p_Y}[-\log p_Y(Y)]$ and the conditional entropy of $Y$ given $Z$ as $\mathcal{H}(Y|Z) := \mathbb{E}_{p_{(Y,Z)}}[-\log p_{Y|Z}(Y|Z)]$. Besides, we define the cross-entropy (CE) between $Y$ and $\hat{Y}$ by $\mathcal{H}(Y; \hat{Y}) := \mathbb{E}_{p_Y}[-\log p_{\hat{Y}}(Y)]$ and the conditional CE given $Z$ by $\mathcal{H}(Y; \hat{Y}|Z) := \mathbb{E}_{p_{(Y,Z)}}[-\log p_{\hat{Y}|Z}(Y|Z)]$. Before our analysis, we first revisit contrastive loss.

**Contrastive loss.** We use the supervised contrastive loss [25] for fine-tuning, which is a variant of InfoNCE [43]. Specifically, given a sample feature $z_i$ as anchor, the contrastive loss takes the features from the same class to the anchor as positive pairs and those from different classes as negative pairs. Assuming features are $\ell_2$-normalized, the contrastive loss is computed by:

$$\mathcal{L}_{con} = -\frac{1}{n} \sum_{i=1}^n \frac{1}{|P_i|} \sum_{z_j \in P_i} \log \frac{e^{(z_i^\top z_j / \tau)}}{\sum_{z_k \in A_i} e^{(z_i^\top z_k / \tau)}}, \tag{1}$$

where $\tau$ is a temperature factor, while $P_i$ and $A_i$ denote the positive pair set and the full pair set of the anchor $z_i$, respectively. We next analyze the contrastive loss and find it has two beneficial effects.

## 3.1 Regularization Effect of Contrastive Loss

We first show the contrastive loss has regularization effectiveness on representation learning based on the following theorem.

**Theorem 1** *Assuming the features are $\ell_2$-normalized and the classes are balanced with equal data number, minimizing the contrastive loss is equivalent to minimizing the class-conditional entropy $\mathcal{H}(Z|Y)$ and maximizing the feature entropy $\mathcal{H}(Z)$:*

$$\mathcal{L}_{con} \propto \mathcal{H}(Z|Y) - \mathcal{H}(Z)$$

Please see Appendix A for the proof. This theorem shows that $\mathcal{L}_{con}$ explicitly regularizes representation learning. On one hand, minimizing $\mathcal{L}_{con}$ will minimize $\mathcal{H}(Z|Y)$, which encourages learning a low-entropy cluster for each class (*i.e.,* high intra-class compactness). On the other hand, minimizing $\mathcal{L}_{con}$ will maximize $\mathcal{H}(Z)$ and tends to learn a high-entropy feature space (*i.e.,* large inter-class separation degree). This provides an additional regularization effect on the feature space, which can be observed by the feature visualization in Figure 2. As for the two assumptions, $\ell_2$-normalized features can be satisfied by a non-linear projection in practice (cf. Section 4.1), while

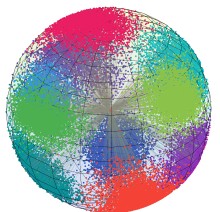 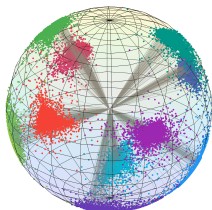

(a) Training with $\mathcal{L}_{ce}$     (b) Training with $\mathcal{L}_{ce}+\mathcal{L}_{con}$

Figure 2: Visualizations of features learned by ResNet-18 on the CIFAR10 validation set. Compared to training with only cross-entropy $\mathcal{L}_{ce}$, the contrastive loss $\mathcal{L}_{con}$ helps to regularize the feature space and make it more discriminative. Best viewed in color.

contrastive fine-tuning also empirically performs well on class-imbalanced datasets (cf. Table 6). Note that this analysis is different from the analysis in unsupervised contrastive learning [53], which is specific to the (unlabeled) instance level rather than the class level.

## 3.2 Optimization Effect of Contrastive Loss

We next show that the contrastive loss improves the optimization effectiveness during model training via Theorem 2.

**Theorem 2** *Assuming the features are $\ell_2$-normalized and the classes are balanced, the contrastive loss is positive proportional to the infimum of conditional cross-entropy $\mathcal{H}(Y; \hat{Y}|Z)$, where the infimum is taken over classifiers:*

$$\mathcal{L}_{con} \propto \inf \underbrace{\mathcal{H}(Y; \hat{Y}|Z)}_{Conditional\ CE} - \mathcal{H}(Y)$$

Please see Appendix A for proofs. This theorem shows $\mathcal{L}_{con}$ boosts model optimization. Concretely, the label $Y$ is given by datasets, so its entropy $\mathcal{H}(Y)$ is a constant and can be ignored. Hence, minimizing $\mathcal{L}_{con}$ will minimize the infimum of conditional cross-entropy $\mathcal{H}(Y; \hat{Y}|Z)$, which provides an additional optimization effect as compared to fine-tuning with only cross-entropy. More intuitively, pulling positive pairs together and pushing negative pairs further apart make the predicted label distribution closer to the ground-truth distribution, which further minimizes the cross-entropy loss.

## 4 Contrast-Regularized Tuning

Based on the above theoretical analysis, we are motivated to introduce contrastive learning to fine-tune contrastive self-supervised visual models on downstream tasks. Nevertheless, we empirically find that simply adding the contrastive loss to the fine-tuning objective is insufficient to obtain promising performance (cf. Table 2). One key cause is that contrastive learning highly relies on positive/negative sample pairs, but the majority of samples are easy-to-contrast pairs [19, 56] that may produce negligible contrastive loss gradients. This makes contrastive learning fail to learn more discriminative representations and thus suffer from unsatisfactory performance. To address this issue and better fine-tune contrastive self-supervised models, we propose a new contrast-regularized tuning (Core-tuning) method based on a novel hard sample pair mining strategy as follows.

## 4.1 Hard Sample Pair Mining for Contrastive Fine-Tuning

For more effective contrastive fine-tuning, Core-tuning generates both hard positive and hard negative pairs via a new hardness-directed mixup strategy, and meanwhile assigns higher importance weights to hard positive pairs via a new focal contrastive loss.

**Hard positive pair generation.** As shown in Figure 1 (b), for a given feature anchor $z_i$, we first find its hardest positive data $(z_i^{hp}, y_i^{hp})$ and hardest negative data $(z_i^{hn}, y_i^{hn})$ based on cosine similarity. That is, $z_i^{hp}$ is the positive data (from the same class) with the lowest cosine similarity to the anchor, and $z_i^{hn}$ is the negative data (from different classes) most similar to the anchor. We then generate a hard positive pair as a convex combination of the two hardest pairs:

$$z_i^+ = \lambda z_i^{hp} + (1-\lambda)z_i^{hn}; \quad y_i^+ = \lambda y_i^{hp} + (1-\lambda)y_i^{hn},$$

where $\lambda \sim \text{Beta}(\alpha, \alpha) \in [0, 1]$ [64], in which $\alpha \in (0, \infty)$ is a hyper-parameter to decide the Beta distribution. The generated positive pairs are located between positives and negatives and thus are harder to contrast. Note that the generated positive pairs do not have to be the hardest. In fact, as long as we can generate relatively hard pairs, the performance of contrastive fine-tuning could be improved.

**Hard negative pair generation.** As shown in Figure 1 (c), for a given feature anchor $z_i$, we randomly select a negative sample $(z_i^n, y_i^n)$ to synthesize a semi-hard negative pair as follows:

$$z_i^- = (1-\lambda)z_i + \lambda z_i^n; \quad y_i^- = (1-\lambda)y_i + \lambda y_i^n,$$

where $\lambda \sim \text{Beta}(\alpha, \alpha)$. The reason why we select a random negative sample instead of the hardest negative is that generating too hard negatives may result in false negatives and degrade performance. Note that semi-hard negatives may even yield better performance in metric learning [57].

**Hard pair reweighting.** After generating hard sample pairs, we use an additional two-layer MLP head $G_c$ to obtain $\ell_2$-normalized contrastive features $v_i = G_c(z_i)/\|G_c(z_i)\|_2$, since a nonlinear projection improves contrastive learning [7, 8]. Based on these features, one may directly use $\mathcal{L}_{con}$ in Eq. (1) for fine-tuning. However, since hard pairs are more informative for contrastive learning, we propose to assign higher importance weights to hard positive pairs. Inspired by focal loss [36], we find that hard positive pairs generally lead to a low prediction probability $p_{ij} = \frac{\exp(v_i^\top v_j/\tau)}{\sum_{v_k \in A_i} \exp(v_i^\top v_k/\tau)}$.

Thus, we reweight $\mathcal{L}_{con}$ with $(1-p_{ij})$ and develop a focal contrastive loss:

$$\mathcal{L}_{con}^f = -\frac{1}{n}\sum_{i=1}^n \frac{1}{|P_i|} \sum_{v_j \in P_i} (1-p_{ij}) \log \frac{e^{(v_i^\top v_j/\tau)}}{\sum_{v_k \in A_i} e^{(v_i^\top v_k/\tau)}},$$

where $P_i$, $A_i$ denote the anchor's positive and full pair sets, which contain the generated hard pairs. Via the hard pair mining strategy, Core-tuning is able to learn a more discriminative feature space.

## 4.2 Overall Training Scheme and Smooth Classifier Learning

In fine-tuning, both the feature extractor and classifier need to be trained, so the final training scheme of Core-tuning[2] is to minimize the following objective:

$$\min \quad \underbrace{\mathcal{L}_{ce}^m}_{\text{cross-entropy loss}} + \underbrace{\eta\mathcal{L}_{con}^f}_{\text{focal contrastive loss}},$$

where $\eta$ is a trade-off factor. Since hard sample mining has helped to learn a discriminative feature space, the remaining question is how to train the classifier for better exploiting such a feature space.

**Smooth classifier learning.** Previous work [52] has found that the classifier simply trained with cross-entropy is often sharp and close to data. This may make the classifier fail to exploit the high inter-class separation degree in the discriminative feature space due to closeness to training data, as well as suffer from limited generalization performance since the classifier near the training data may lead to incorrect yet confident predictions when evaluated on slightly different test samples. To address this, inspired by the effectiveness of mixup for helping learn a smoother decision boundary [40, 52], we further use the mixed data from the generated hard sample pair set (denoted by $\mathcal{B}$) for classifier training: $\mathcal{L}_{ce}^m = -\frac{1}{n}\sum_{i=1}^n y_i \log(\hat{y}_i) - \frac{1}{|\mathcal{B}|}\sum_{(z_j, y_j) \in \mathcal{B}} y_j \log(G_y(z_j))$. In this way, Core-tuning is able to learn a smoother classifier that is far away from the training data, and thus can better exploit the learned discriminative feature space and improve the model generalizability.

---

[2]The pseudo code is provided in the supplementary.

Table 1: Comparisons of various fine-tuning methods for the MoCo-v2 pre-trained ResNet-50 model on image classification in terms of top-1 accuracy. SL-CE-tuning denotes supervised pre-training on ImageNet and then fine-tuning with cross-entropy.

| Method | ImageNet20 | CIFAR10 | CIFAR100 | Caltech101 | DTD | Aircraft | Cars | Pets | Flowers | Avg. |
|---|---|---|---|---|---|---|---|---|---|---|
| SL-CE-tuning | 91.01 | 94.23 | 83.40 | 93.39 | 74.40 | 87.03 | 89.77 | 92.17 | 98.78 | 89.35 |
| CE-tuning | 88.28 | 94.70 | 80.27 | 91.87 | 71.68 | 86.87 | 88.61 | 89.05 | 98.49 | 87.76 |
| L2SP [35] | 88.49 | 95.14 | 81.43 | 91.98 | 72.18 | 86.55 | 89.00 | 89.43 | 98.66 | 88.10 |
| M&M [62] | 88.53 | 95.02 | 80.58 | 92.91 | 72.43 | 87.45 | 88.90 | 89.60 | 98.57 | 88.22 |
| DELTA [33] | 88.35 | 94.76 | 80.39 | 92.19 | 72.23 | 87.05 | 88.73 | 89.54 | 98.65 | 87.99 |
| BSS [9] | 88.34 | 94.84 | 80.40 | 91.95 | 72.22 | 87.18 | 88.50 | 89.50 | 98.57 | 87.94 |
| RIFLE [34] | 89.06 | 94.71 | 80.36 | 91.94 | 72.45 | 87.60 | 89.72 | 90.05 | 98.70 | 88.29 |
| SCL [18] | 89.29 | 95.33 | 81.49 | 92.84 | 72.73 | 87.44 | 89.37 | 89.71 | 98.65 | 88.54 |
| Bi-tuning [71] | 89.06 | 95.12 | 81.42 | 92.83 | 73.53 | 87.39 | 89.41 | 89.90 | 98.57 | 88.58 |
| Core-tuning (ours) | **92.73** | **97.31** | **84.13** | **93.46** | **75.37** | **89.48** | **90.17** | **92.36** | **99.18** | **90.47** |

## 5  Experiments

We first test the effectiveness of Core-tuning on image classification and then apply it to semantic segmentation. Next, since Core-tuning potentially improves model generalizability, we further study how it affects model generalization to new domains and model robustness to adversarial samples.

### 5.1  Results on Image Classification

**Settings.** As there is no fine-tuning method devoted to contrastive self-supervised models, we compare Core-tuning with advanced fine-tuning methods for general models (*e.g.,* supervised pre-trained models): L2SP [35], M&M [62], DELTA [33], BSS [9], RIFLE [34], SCL [18] and Bi-tuning [71]. We denote the fine-tuning with cross-entropy by CE-tuning.

Following [27], we test on 9 natural image datasets, including ImageNet20 (a subset of ImageNet with 20 classes), CIFAR10, CIFAR100 [29], Caltech-101 [15], DTD [10], FGVC Aircraft [39], Standard Cars [28], Oxford-IIIT Pets [44] and Oxford 102 Flowers [42]. Specifically, ImageNet20 is an ImageNet subset with 20 classes, by combining the ImageNette and ImageWoof datasets [21]. Here, we do not directly test on ImageNet [11], since all CSL models are pre-trained on the ImageNet dataset. These datasets cover a wide range of fine/coarse-grained object recognition tasks.

We implement Core-tuning in PyTorch[3]. Following [13], we use ResNet-50 ($1\times$), pre-trained by various CSL methods on ImageNet, as the network backbone. All checkpoints of pre-trained models are provided by authors or by the PyContrast repository[4]. Following [6], we perform parameter tuning for $\eta$ and $\alpha$ from $\{0.1, 1, 10\}$ on each dataset. Moreover, we set the temperature $\tau{=}0.07$. To make the generated negative pairs closer to negatives, we clip $\lambda{\sim}\text{Beta}(\alpha, \alpha)$ by $\lambda{\geq}\lambda_n$ when generating hard negative pairs, where $\lambda_n$ is a threshold and we set it to 0.8. All results are averaged over 3 runs in terms of the top-1 accuracy. More dataset details, more implementation details and the parameter analysis are put in Appendices C and E.

**Comparisons with previous methods.** We report the fine-tuning performance of the MoCo-v2 pre-trained model in Table 1. When using the standard CE-tuning, the MoCo-v2 pre-trained model performs worse than the supervised pre-trained model on most datasets. This is because the self-supervised pre-trained model is less class-discriminative than the supervised pre-trained model due to the lack of annotations during pre-training. Moreover, the classic fine-tuning methods designed for supervised pre-trained models (*e.g.,* L2SP and DELTA) cannot fine-tune the contrastive self-supervised model very well. One reason is that the contrastive pre-training task is essentially different from the downstream classification task, so strictly regularizing the difference between the contrastive self-supervised model and the fine-tuned model may lead to negative/poor transfer. In addition, M&M, SCL and Bi-tuning use the triplet loss or the contrastive loss during fine-tuning. However, they ignore the two challenges in contrastive fine-tuning as mentioned in Figure 1, leading to limited model performance on downstream tasks. In contrast, Core-tuning handles those challenges well and improves the fine-tuning performance of CSL models a lot. This result demonstrates the superiority of Core-tuning. More results like the standard error are put in Appendix D.

---

[3]The source code of Core-tuning is available at: `https://github.com/Vanint/Core-tuning`.
[4]`https://github.com/HobbitLong/PyContrast`.

Table 2: Ablation studies of Core-tuning (Row 5) for fine-tuning MoCo-v2 pre-trained ResNet-50 in terms of top-1 accuracy, where cross-entropy is used in all baselines. Here, $\mathcal{L}_{con}$ is the original contrastive loss, while $\mathcal{L}_{con}^{f}$ is our focal contrastive loss. Moreover, "mix" denotes the manifold mixup, while "mix-H" indicates the proposed hardness-directed mixup strategy in our method.

| $\mathcal{L}_{con}$ | $\mathcal{L}_{con}^{f}$ | mix | mix-H | ImageNet20 | CIFAR10 | CIFAR100 | Caltech101 | DTD | Aircraft | Cars | Pets | Flowers | Avg. |
|---|---|---|---|---|---|---|---|---|---|---|---|---|---|
| | | | | 88.28 | 94.70 | 80.27 | 91.87 | 71.68 | 86.87 | 88.61 | 89.05 | 98.49 | 87.76 |
| √ | | | | 89.29 | 95.33 | 81.49 | 92.84 | 72.73 | 87.44 | 89.37 | 89.71 | 98.65 | 88.54 |
| | | √ | | 90.67 | 95.43 | 81.03 | 92.68 | 73.31 | 88.37 | 89.06 | 91.37 | 98.74 | 88.96 |
| √ | | | √ | 92.20 | 97.01 | 83.89 | 93.22 | 74.78 | 88.88 | 89.79 | 91.95 | 98.94 | 90.07 |
| | √ | | √ | **92.73** | **97.31** | **84.13** | **93.46** | **75.37** | **89.48** | **90.17** | **92.36** | **99.18** | **90.47** |

Table 3: Fine-tuning results of ResNet-50, pre-trained by various methods. "Cont." indicates contrastive self-supervised pre-training; CE indicates cross-entropy.

| Pre-training | Types | Caltech101 | | DTD | | Pets | |
|---|---|---|---|---|---|---|---|
| | | CE | ours | CE | ours | CE | ours |
| InsDis [58] | | 82.30 | **88.60** | 69.81 | **70.94** | 87.57 | **89.59** |
| PIRL [41] | Cont. | 84.23 | **89.29** | 68.95 | **71.72** | 86.87 | **89.52** |
| MoCo-v1 [20] | | 85.74 | **89.16** | 69.91 | **71.90** | 88.16 | **90.11** |
| InfoMin [47] | | 92.73 | **94.01** | 72.59 | **74.89** | 90.00 | **92.34** |
| DeepCluster[2] | | 89.99 | **92.34** | 72.77 | **75.21** | 90.53 | **93.17** |
| SwAV [3] | Non-Cont. | 87.71 | **91.34** | 75.29 | **77.41** | 92.48 | **93.29** |
| BYOL [17] | | 91.19 | **93.25** | 74.94 | **76.56** | 92.39 | **93.74** |
| CE | Supervised | 93.65 | **94.20** | 74.40 | **77.27** | 92.17 | **93.82** |

Table 4: Fine-tuning performance of various architectures. Here, ResNet (R) and ResNeXt (RX) are pre-trained by InfoMin; DeiT-S [48] is pre-trained by DINO [4].

| Archs. | Caltech101 | | DTD | | Pets | |
|---|---|---|---|---|---|---|
| | CE | ours | CE | ours | CE | ours |
| R-50 | 92.73 | **94.01** | 72.59 | **74.89** | 90.00 | **92.34** |
| R-101 | 93.06 | **94.33** | 73.38 | **75.09** | 90.84 | **92.91** |
| R-152 | 93.39 | **94.66** | 73.74 | **75.42** | 91.08 | **92.97** |
| RX-101 | 93.71 | **95.12** | 74.43 | **75.97** | 91.97 | **94.04** |
| RX-152 | 93.92 | **95.19** | 74.76 | **76.22** | 92.70 | **94.49** |
| DeiT-S/16 | 91.24 | **92.31** | 71.35 | **72.83** | 92.43 | **93.72** |

**Ablation studies of Core-tuning.** We conduct ablation studies for Core-tuning regarding the focal contrastive loss and the hardness-directed mixup strategy. As shown in Table 2, each component improves the fine-tuning performance in Core-tuning. Note that the mixup in Row 3 is the manifold mixup [52], which is essentially designed for classification and is expected to outperform our hardness-directed mixup strategy regarding classification performance. However, our proposed Core-tuning (Row 5) still shows obvious improvement on all datasets, which strongly verifies the value of contrastive fine-tuning. More ablation results for verifying the effectiveness of hard pair generation and smooth classifier learning are put in Appendix E.

**Results on different pre-training methods.** In previous experiments, we fine-tune the MoCo-v2 pre-trained ResNet-50, but it is unclear whether Core-tuning can be applied to fine-tune models with other pre-training methods. Hence, we further use Core-tuning to fine-tune ResNet-50, pre-trained by other CSL methods (*i.e.,* InsDis [58], PIRL [41], MoCo-v1 [20] and InfoMin [47]), non-contrastive self-supervised methods (*i.e.,* DeepCluster-v2 [2], SwAV [3] and BYOL [17]), and supervised learning. As shown in Table 3, Core-tuning fine-tunes all pre-trained models consistently better than CE-tuning on 3 image classification datasets. Such results verify the generalizability of the proposed Core-tuning. More results on different pre-trained models are put in Appendix D.

**Results on different network architectures.** Previous experiments are based on ResNet-50, while it is unclear whether Core-tuning can be applied to other network architectures. Hence, we further use Core-tuning to fine-tune various residual network architectures (*i.e.,* ResNet-101 and 152; ResNeXt-101 and 152 [59]) pre-trained by InfoMin [47], and vision transformer (*i.e.,* DeiT-S/16 [48]) pre-trained by DINO [4]. As shown in Table 4, Core-tuning fine-tunes all network architectures well on all three datasets, showing strong universality.

**Results on different data sizes.** The labeled data may be scarce in downstream tasks. Hence, we further evaluate Core-tuning on ImageNet20 with different sampling rates of data. We report the results in Table 5, while the results on the full ImageNet20 have been listed in Table 1. Specifically, Core-tuning outperforms baselines in all cases. Note that when the data is very scarce (*e.g.,* 10%), the fine-tuning performance of CE-tuning degrades and fluctuates significantly, in which case Core-tuning obtains more significant improvement and achieves more stable performance.

**Results on large-scale and class-imbalanced dataset.** The real-world datasets may be large-scale and class-imbalanced [67, 68, 70], so we also evaluate Core-tuning on a long-tailed iNaturalist18 dataset [50], consisting 437,513 images from 8,142 classes. As shown in Table 6, Core-tuning also performs well on the large-scale and class-imbalanced dataset for fine-tuning contrastive pre-trained models. Note that in our theoretical analysis, we assume that the classes are balanced with the same data number to facilitate analysis. Nevertheless, this assumption does not mean that contrastive fine-tuning cannot handle class-imbalanced datasets. Here, the promising results on iNaturalist-18 verify the effectiveness of Core-tuning on highly class-imbalanced scenarios.

Table 5: Fine-tuning performance of the MoCo-v2 pre-trained ResNet-50 with various numbers of labeled data.

| Method | Sampling Rates on ImageNet20 | | | |
|--------|------|------|------|------|
| | 10% | 25% | 50% | 75% |
| CE-tuning | 52.97+/-3.96 | 63.17+/-3.94 | 81.78+/-1.37 | 85.85+/-0.11 |
| Bi-tuning | 60.50+/-1.11 | 75.86+/-0.74 | 83.18+/-0.27 | 87.19+/-0.19 |
| Core-tuning | **78.64+/-0.58** | **84.48+/-0.34** | **89.09+/-0.40** | **90.93+/-0.24** |

Table 6: Fine-tuning performance of the MoCo-v2 pre-trained ResNet-50 on large-scale and class-imbalanced iNaturalist18 in terms of top-1 accuracy.

| Fine-tuning method | iNaturalist18 |
|--------------------|---------------|
| CE-tuning | 61.72+/-0.18 |
| CE-Contrastive-tuning | 62.75+/-0.22 |
| Core-tuning (ours) | **63.57+/-0.09** |

Table 7: Fine-tuning performance on PASCAL VOC semantic segmentation based on DeepLab-V3 with ResNet-50, pre-trained by various CSL methods.

| Pre-training | Fine-tuning | MPA | FWIoU | MIoU |
|--------------|-------------|-----|-------|------|
| Supervised | CE | 87.10 | 89.12 | 76.52 |
| InsDis | CE | 83.64 | 88.23 | 74.14 |
| | ours | **84.53** | **88.67** | **74.81** |
| PIRL | CE | 83.16 | 88.22 | 73.99 |
| | ours | **85.30** | **88.95** | **75.49** |
| MoCo-v1 | CE | 84.71 | 88.75 | 74.94 |
| | ours | **85.70** | **89.19** | **75.94** |
| MoCo-v2 | CE | 87.31 | 90.26 | 78.42 |
| | ours | **88.76** | **90.75** | **79.62** |
| InfoMin | CE | 87.17 | 89.84 | 77.84 |
| | ours | **88.92** | **90.65** | **79.48** |

## 5.2 Results on Semantic Segmentation

We next apply Core-tuning to fine-tune contrastive self-supervised models on semantic segmentation.

**Implementation details.** We adopt the DeepLab-V3 framework [5] for PASCAL VOC semantic segmentation and use CSL pre-trained ResNet-50 models as the backbone. In Core-tuning, we enforce the contrastive regularizer after the penultimate layer of ResNet-50 via an additional global average pooling. Following [54], the model is fine-tuned on VOC train_aug2012 set for 30k steps via SGD based on two GPUs and evaluated on val2012 set. The image is rescaled to $513 \times 513$ with random crop and flips for training and with center crop for evaluation. The batch size and output stride are 16. Besides, we set the initial learning rate to 0.1 and adjust it via the poly decay schedule. Other parameters are the same as image classification. We use three metrics: Mean Pixel Accuracy (MPA), Frequency Weighted Intersection over Union (FWIoU) and Mean Intersection over Union (MIoU).

**Results.** As shown in Table 7, Core-tuning contributes to the fine-tuning performance of all CSL models in terms of MPA, FWIoU and MIoU. The promising results demonstrate the effectiveness of Core-tuning on semantic segmentation. Interestingly, we find that with standard fine-tuning, the models pre-trained by MoCo-v2 and InfoMin have already outperformed the supervised pre-trained model. One explanation is that self-supervised pre-training may keep more visual information, compared to supervised pre-training that mainly extracts information specific to classification [69]. In other words, unsupervised contrastive learning may extract more beneficial information for dense prediction, which inspires us to explore unsupervised contrastive regularizers in the future.

## 5.3 Effectiveness on Cross-Domain Generalization

The generalizability of deep networks to unseen domains is important for their application to real-world scenarios [12]. We thus wonder whether Core-tuning also benefits model generalization on downstream tasks, so we apply Core-tuning to the task of domain generalization (DG).

**Implementation details.** DG aims to train a model on multiple source domains and expect it to generalize well to an unseen target domain. Specifically, we use MoCo-v2 pre-trained ResNet-50 as the backbone, and evaluate Core-tuning on 3 benchmark datasets, *i.e.,* PACS [32], VLCS [14] and Office-Home [51]. For training, we use Adam optimizer with batch size 32. The learning rate is set to $5 \times 10^{-5}$ and the training step is 20,000. More implementation details are put in Appendix C.

**Results.** We report the results on PACS and VLCS in Table 8 and the results on OfficeHome in Appendix D, from which we draw observations as follows. First, when fine-tuning with cross-entropy, the contrastive self-supervised model performs worse than the supervised pre-trained model. This results from the relatively worse discriminative abilities of the contrastive self-supervised model, which can also be found in Table 1. Second, enforcing the contrastive regularizer during fine-tuning improves DG performance, since the contrastive regularizer helps to learn more discriminative features (cf. Theorem 1) and also helps to alleviate distribution shifts among domains [24]. Last, Core-tuning further improves the generalization performance of models. This is because hard pair generation further boosts contrastive learning, while smooth classifier learning benefits model generalizability. We thus conclude that Core-tuning is beneficial to model generalization on downstream tasks.

Table 8: Domain generalization accuracies of various fine-tuning methods for MoCo-v2 pre-trained ResNet-50. CE means cross-entropy; CE-Con enhances CE with the contrastive loss. Moreover, A/C/P/S and C/L/V/S are different domains in PACS and VLCS datasets, respectively.

| Pre-training | Fine-tuning | PACS | | | | | VLCS | | | | |
|---|---|---|---|---|---|---|---|---|---|---|---|
| | | A | C | P | S | Avg. | C | L | V | S | Avg. |
| Supervised | CE | 83.65 | 79.21 | 96.11 | 81.46 | 85.11 | 98.41 | 63.81 | 68.55 | 75.45 | 76.56 |
| MoCo-v2 | CE | 78.71 | 76.92 | 90.87 | 75.67 | 80.54 | 94.96 | 66.87 | 68.96 | 64.98 | 73.94 |
| | CE-Con | 85.11 | 81.77 | 95.58 | 80.12 | 85.65 | 95.94 | 67.76 | 69.31 | 73.57 | 77.67 |
| | ours | **87.31** | **84.06** | **97.53** | **83.43** | **88.08** | **98.50** | **68.19** | **73.15** | **81.53** | **80.34** |

Table 9: Adversarial training performance of MoCo-v2 pre-trained ResNet-50 on CIFAR10 under the attack of PGD-10 in terms of robust and clean accuracies. AT-CE indicates adversarial training (AT) with CE; AT-CE-Con enhances AT-CE with the contrastive loss; AT-ours uses Core-tuning for AT.

| Method | $\ell_2$-attack | | | | | | $\ell_\infty$-attack | | | | | |
|---|---|---|---|---|---|---|---|---|---|---|---|---|
| | $\epsilon$=0.5 | | $\epsilon$=1.5 | | $\epsilon$=2.5 | | $\epsilon$= 2/255 | | $\epsilon$= 4/255 | | $\epsilon$= 8/255 | |
| | Robust | Clean | Robust | Clean | Robust | Clean | Robust | Clean | Robust | Clean | Robust | Clean |
| CE | 50.25 | 94.70 | 48.29 | 94.70 | 46.82 | 94.70 | 25.13 | 94.70 | 12.28 | 94.70 | 4.57 | 94.70 |
| AT-CE | 86.59 | 92.00 | 89.60 | 94.28 | 89.16 | 94.15 | 83.20 | 93.05 | 75.82 | 91.99 | 69.27 | 92.79 |
| AT-CE-Con | 90.74 | 94.71 | 90.29 | 94.80 | 89.70 | 94.27 | 85.07 | 94.56 | 79.75 | 93.79 | 70.70 | 93.38 |
| AT-ours | **92.97** | **96.82** | **92.32** | **96.90** | **92.05** | **96.87** | **86.92** | **96.29** | **82.01** | **95.95** | **74.83** | **95.90** |

## 5.4 Robustness to Adversarial Samples

As is known, deep networks are fragile to adversarial attack [46]. We next study whether Core-tuning also benefits model robustness to adversarial samples in the setting of adversarial training (AT).

**Implementation details.** We use MoCo-v2 pre-trained ResNet-50 as the network backbone, and use the Projected Gradient Descent (PGD) [38] to generate adversarial samples with $\ell_2$ attack (strength $\sigma$=0.5) and $\ell_\infty$ attack (strength $\sigma$=4/255). During AT, we use both original samples and adversarial samples for fine-tuning. Moreover, we use the clean accuracy on original samples and the robust accuracy on adversarial samples as metrics. More implementation details are put in Appendix C.

**Results.** We report the results on CIFAR10 in Table 9 and the results on Caltech-101, DTD and Pets in Appendix D. First, despite good clean accuracy, fine-tuning with cross-entropy cannot defend against adversarial attack, leading to poor robust accuracy. Second, AT with cross-entropy improves the robust accuracy significantly, but it inevitably degrades the clean accuracy due to the well-known accuracy-robustness trade-off [49]. In contrast, the contrastive regularizer improves both robust and clean accuracies. This is because contrastive learning helps to improve robustness generalization (*i.e.,* alleviating the distribution shifts between clean and adversarial samples). Last, Core-tuning further boosts AT and, surprisingly, even achieves better clean accuracy than the standard fine-tuning with cross-entropy. To our knowledge, this is quite promising since even the most advanced AT methods [61, 65] find it difficult to overcome the accuracy-robustness trade-off [63]. The improvement is because both contrastive learning and smooth classifier learning boost robustness generalization. We thus conclude that Core-tuning improves model robustness on downstream tasks.

## 6 Conclusions

This paper studies how to fine-tune contrastive self-supervised visual models. We theoretically show that optimizing the contrastive loss during fine-tuning has regularization effectiveness on representation learning as well as optimization effectiveness on classifier training, both of which benefit model fine-tuning. We thus propose a novel contrast-regularized tuning (Core-tuning) method to fine-tune CSL visual models. Promising results on image classification and semantic segmentation verify the effectiveness of Core-tuning. Also, we empirically find that Core-tuning is beneficial to model generalization and robustness on downstream tasks. We thus recommend using Core-tuning as a standard baseline to fine-tune CSL visual models, and also call for more attention to the fine-tuning of CSL visual models on understanding its underlying theories and better approaches in the future.

**Limitation discussion.** One potential limitation of Core-tuning is that it is specifically designed for and also focuses on the fine-tuning of CSL visual models. Considering the universality of Core-tuning (cf. Table 3), we will explore the extension of Core-tuning to better fine-tune supervised pre-trained and other self-supervised visual models and even language models on more tasks.

## Acknowledgments

This work was partially supported by NUS ODPRT Grant R252-000-A81-133.

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
