# Supplementary Material:
# Unleashing the Power of Contrastive Self-Supervised Visual Models via Contrast-Regularized Fine-Tuning

**Yifan Zhang**[1][*]   **Bryan Hooi**[1]   **Dapeng Hu**[1]   **Jian Liang**[2]   **Jiashi Feng**[3]
[1]National University of Singapore   [2]Chinese Academy of Sciences   [3]SEA AI Lab

## Abstract

We provide the supplementary material for our paper "Unleashing the Power of Contrastive Self-Supervised Visual Models via Contrast-Regularized Fine-Tuning" [**?**], including proofs for the analysis of the contrastive loss (cf. Appendix A), the pseudo-code of the proposed method (cf. Appendix B), more implementation details (cf. Appendix C), and more empirical results and analysis (cf. Appendix D and Appendix E).

## A  Proof of Theoretical Analysis

This appendix provides proofs for both Theorems 1 and 2.

### A.1  Proof for Theorem 1

**Theorem 1** *Assuming the features are $\ell_2$-normalized and the classes are balanced with equal data number, minimizing the contrastive loss is equivalent to minimizing the class-conditional entropy $\mathcal{H}(Z|Y)$ and maximizing the feature entropy $\mathcal{H}(Z)$:*

$$\mathcal{L}_{con} \ \propto \ \mathcal{H}(Z|Y) \ - \ \mathcal{H}(Z)$$

**Proof**   We follow the notations in the main paper and further denote the sample set of the class $k$ by $\mathcal{Z}_k$. Moreover, we assume the classes of samples are balanced so that the sample number of each class is constant $|\mathcal{Z}_k| = \frac{n}{K}$, where $n$ denotes the total number of samples and $K$ indicates the number of classes. Let us start by splitting the contrastive loss into two terms.

$$
\begin{aligned}
\mathcal{L}_{con} &= -\frac{1}{n}\sum_{i=1}^{n}\frac{1}{|P_i|}\sum_{z_j \in P_i}\log\frac{e^{(v_i^\top v_j/\tau)}}{\sum_{v_k \in A_i}e^{(v_i^\top v_k/\tau)}} \\
&= -\frac{1}{n}\sum_{i=1}^{n}\frac{1}{|P_i|}\sum_{z_j \in P_i}\frac{z_i^\top z_j}{\tau}+\frac{1}{n}\sum_{i=1}^{n}\log\sum_{z_k \in A_i}e^{(\frac{z_i^\top z_k}{\tau})}.
\end{aligned}
\tag{1}
$$

Let $c_k = \frac{1}{|\mathcal{Z}_k|}\sum_{z \in \mathcal{Z}_k} z$ denote the hard mean of all features from the class $k$, and let the symbol $\overset{c}{=}$ indicate equality up to a multiplicative and/or additive constant. We first analyze the first term in

---
[*]Corresponding to: Yifan Zhang <yifan.zhang@u.nus.edu>

35th Conference on Neural Information Processing Systems (NeurIPS 2021).

Eq. (1) by connecting it to a tightness term of the center loss, *i.e.,* $\sum_{z_i \in \mathcal{Z}_k} \|z_i - c_k\|^2$ [55]:

$$
\sum_{z_i, z_j \in \mathcal{Z}_k} -\frac{z_i^\top z_j}{\tau} \overset{c}{=} \frac{1}{|\mathcal{Z}_k|} \sum_{z_i, z_j \in \mathcal{Z}_k} -z_i^\top z_j
$$

$$
\overset{c}{=} \frac{1}{|\mathcal{Z}_k|} \sum_{z_i, z_j \in \mathcal{Z}_k} \|z_i\|^2 - z_i^\top z_j
$$

$$
= \sum_{z_i \in \mathcal{Z}_k} \|z_i\|^2 - \frac{1}{|\mathcal{Z}_k|} \sum_{z_i \in \mathcal{Z}_k} \sum_{z_j \in \mathcal{Z}_k} z_i^\top z_j
$$

$$
= \sum_{z_i \in \mathcal{Z}_k} \|z_i\|^2 - 2\frac{1}{|\mathcal{Z}_k|} \sum_{z_i \in \mathcal{Z}_k} \sum_{z_j \in \mathcal{Z}_k} z_i^\top z_j
$$

$$
+ \frac{1}{|\mathcal{Z}_k|} \sum_{z_i \in \mathcal{Z}_k} \sum_{z_j \in \mathcal{Z}_k} z_i^\top z_j
$$

$$
= \sum_{z_i \in \mathcal{Z}_k} \|z_i\|^2 - 2z_i^\top c_k + \|c_k\|^2
$$

$$
= \sum_{z_i \in \mathcal{Z}_k} \|z_i - c_k\|^2,
$$

where we use the property of $\ell_2$-normalized features that $\|z_i\|^2 = \|z_j\|^2 = 1$ and the definition of the class hard mean $c_k = \frac{1}{|\mathcal{Z}_k|} \sum_{z \in \mathcal{Z}_k} z$.

By summing over all classes $k$, we obtain:

$$
\sum_{i=1}^{n} \sum_{z_j \in P_i} -\frac{z_i^\top z_j}{\tau} \overset{c}{=} \sum_{i=1}^{n} \|z_i - c_{y_i}\|^2.
$$

Based on this equation, following [1], we can interpret the first term in Eq. (1) as a conditional cross-entropy between $Z$ and another random variable $\bar{Z}$, whose conditional distribution given $Y$ is a standard Gaussian centered around $c_Y : \bar{Z}|Y \sim \mathcal{N}(c_y, i)$:

$$
-\frac{1}{n} \sum_{i=1}^{n} \frac{1}{|P_i|} \sum_{z_j \in P_i} \frac{z_i^\top z_j}{\tau} \overset{c}{=} \mathcal{H}(Z; \bar{Z}|Y) = \mathcal{H}(Z|Y) + \mathcal{D}_{KL}(Z||\bar{Z}|Y).
$$

Based on this, we know that the first term in Eq. (1) is an upper bound on the conditional entropy of features $Z$ given labels $Y$:

$$
-\frac{1}{n} \sum_{i=1}^{n} \frac{1}{|P_i|} \sum_{z_j \in P_i} \frac{z_i^\top z_j}{\tau} \overset{c}{\geq} \mathcal{H}(Z|Y).
$$

where the symbol $\overset{c}{\geq}$ indicates "larger than" up to a multiplicative and/or an additive constant. When $Z|Y \sim \mathcal{N}(c_y, i)$, the bound is tight. As a result, minimizing the first term in Eq. (1) is equivalent to minimizing $\mathcal{H}(Z|Y)$:

$$
-\frac{1}{n} \sum_{i=1}^{n} \frac{1}{|P_i|} \sum_{z_j \in P_i} \frac{z_i^\top z_j}{\tau} \propto \mathcal{H}(Z|Y). \tag{2}
$$

This concludes the proof for the relationship of the first term in Eq. (1).

We then analyze the second term in Eq. (1), which has the following relationship:

$$\frac{1}{n}\sum_{i=1}^{n}\log\sum_{z_k\in A_i}e^{(\frac{z_i^\top z_k}{\tau})}$$

$$=\frac{1}{n}\sum_{i=1}^{n}\log\left(\sum_{k:y_i=y_k}e^{(\frac{z_i^\top z_k}{\tau})}+\sum_{k:y_i\neq y_k}e^{(\frac{z_i^\top z_k}{\tau})}\right)$$

$$\geq\frac{1}{n}\sum_{i=1}^{n}\log\left(\sum_{k:y_i\neq y_k}e^{(\frac{z_i^\top z_k}{\tau})}\right)$$

$$\overset{c}{\geq}\frac{1}{n}\sum_{i=1}^{n}\sum_{k:y_i\neq y_k}\frac{z_i^\top z_k}{\tau}$$

$$=\frac{1}{n}\sum_{i=1}^{n}\sum_{k=1}^{n}\frac{z_i^\top z_k}{\tau}-\frac{1}{n}\sum_{i=1}^{n}\sum_{k:y_i=y_k}\frac{z_i^\top z_k}{\tau}$$

$$\overset{c}{=}-\frac{1}{n}\sum_{i=1}^{n}\sum_{k=1}^{n}\|z_i-z_k\|^2-\frac{1}{n}\sum_{i=1}^{n}\sum_{k:y_i=y_k}\frac{z_i^\top z_k}{\tau}, \tag{3}$$

where we use Jensen's inequality in the fourth line. The first term in Eq. (3) is close to the differential entropy estimator of features $Z$ provided by [? ]:

$$\hat{\mathcal{H}}(Z)=\frac{d}{n(n-1)}\sum_{i=1}^{n}\sum_{k=1}^{n}\log\|z_i-z_k\|^2\overset{c}{=}\frac{1}{n}\sum_{i=1}^{n}\sum_{k=1}^{n}\log\|z_i-z_k\|^2\propto\frac{1}{n}\sum_{i=1}^{n}\sum_{k=1}^{n}\|z_i-z_k\|^2, \tag{4}$$

where $d$ is the dimension of features. Combining Eq. (3) and Eq. (4) leads to:

$$\frac{1}{n}\sum_{i=1}^{n}\log\sum_{z_k\in A_i}e^{(\frac{z_i^\top z_k}{\tau})}\overset{c}{\geq}-\mathcal{H}(Z)-\frac{1}{n}\sum_{i=1}^{n}\sum_{k:y_i=y_k}\frac{z_i^\top z_k}{\tau}. \tag{5}$$

The second term in the right side of Eq. (5) is essentially a redundant term with the first term in Eq. (1), so we ignore it here. Then, we know that minimizing the second term in Eq. (1) is equivalent to maximizing $\mathcal{H}(Z)$:

$$\frac{1}{n}\sum_{i=1}^{n}\log\sum_{z_k\in A_i}e^{(\frac{z_i^\top z_k}{\tau})}\propto-\mathcal{H}(Z). \tag{6}$$

Combining Eq. (2) and Eq. (6), we conclude the proof of Theorem 1. $\qquad\square$

## A.2 Proof for Theorem 2

**Theorem 2** *Assuming the features are $\ell_2$-normalized and the classes are balanced, the contrastive loss is positive proportional to the infimum of conditional cross-entropy $\mathcal{H}(Y;\hat{Y}|Z)$, where the infimum is taken over classifiers:*

$$\mathcal{L}_{con}\ \propto\ \inf\underbrace{\mathcal{H}(Y;\hat{Y}|Z)}_{Conditional\ CE}\ -\ \mathcal{H}(Y)$$

**Proof** The mutual information between features $Z$ and labels $Y$ can be defined in two ways:

$$\mathcal{I}(Z;Y)=\mathcal{H}(Y)-\mathcal{H}(Y|Z)=\mathcal{H}(Z)-\mathcal{H}(Z|Y). \tag{7}$$

Based on Theorem 1, we know that:

$$\mathcal{L}_{con}\propto\mathcal{H}(Z|Y)-\mathcal{H}(Z)=-\mathcal{I}(Z;Y). \tag{8}$$

Combining Eq. (7) and Eq. (8), we have:

$$\mathcal{L}_{con}\propto\mathcal{H}(Y|Z)-\mathcal{H}(Y). \tag{9}$$

Then, we relate the conditional entropy $\mathcal{H}(Y|Z)$ to the cross entropy loss:

$$\mathcal{H}(Y;\hat{Y}|Z) = \mathcal{H}(Y|Z) + \mathcal{D}_{KL}(Y\|\hat{Y}|Z). \tag{10}$$

According to Eq. (10), when we minimize cross-entropy $\mathcal{H}(Y;\hat{Y}|Z)$, we implicitly minimize both $\mathcal{H}(Y|Z)$ and $\mathcal{D}_{KL}(Y\|\hat{Y}|Z)$. In fact, the optimization could be decoupled into 2 steps in a maximize-minimize (or bound-optimization) way [1]. The first step fixes the parameters of the network encoder and only minimizes Eq. (10) with respect to the parameters of the network classifier. As this step, $\mathcal{H}(Y|Z)$ is fixed and the predictions $\hat{Y}$ are adjusted to minimize $\mathcal{D}_{KL}(Y\|\hat{Y}|Z)$. Ideally, $\mathcal{D}_{KL}(Y\|\hat{Y}|Z)$ would vanish at the end of this step [1]. In this sense, we know that:

$$\mathcal{H}(Y|Z) = \inf \mathcal{H}(Y;\hat{Y}|Z). \tag{11}$$

The second step fixes the classifier and minimizes Eq. (10) with respect to the encoder. By combining Eq. (9) and Eq. (11), we conclude the proof of Theorem 2. $\qquad\square$

## B  Pseudo-code of Core-tuning

We summarize the scheme of Core-tuning in Algorithm 1. Here, all hard pair generation is conducted within each sample batch.

---

**Algorithm 1** The training scheme of Core-tuning.

---

**Require:** Pre-trained encoder $G_e$; Loss factor $\eta$; Mixup factor $\alpha$; Batch size $B$; Epoch number $T$.
**Ensure:** Classifier $G_y$; Projection head $G_c$.
1: **for** t=1,...,T **do**
2:     Sample a batch of training data $\{(x_i, y_i)\}_{i=1}^{B}$;
3:     Obtain features $z_i = G_e(x_i)$ for each sample;
4:     **for** i=1,...,B **do**
5:         Construct positive pair set $P_i$ and full pair set $A_i$ for $z_i$;
6:         Generate hard positive pair $(z_i^+, y_i^+)$ and add it to $P_i$, $A_i$;
7:         Generate hard negative pair $(z_i^-, y_i^-)$ and add it to $A_i$;
8:     **end for**
9:     Obtain contrastive features $v_i = G_c(z_i)$ for all features;
10:    Compute the focal contrastive loss $\mathcal{L}_{con}^{f}$;
11:    Predict $\hat{y}_i = G_y(z_i)$ for the original and generated samples;
12:    Compute the cross-entropy loss $\mathcal{L}_{ce}^{m}$;
13:    loss.backward();    // loss $= \mathcal{L}_{ce}^{m} + \eta\mathcal{L}_{con}^{f}$.
14: **end for**

---

# C  More Experimental Details

## C.1  Implementation Details of Feature Visualization

In the feature visualization, we train ResNet-18 on CIFAR10 with two kinds of losses, *i.e.,* (1) cross-entropy $\mathcal{L}_{ce}$; (2) cross-entropy and the contrastive loss $\mathcal{L}_{ce}+\mathcal{L}_{con}$. For better visualization, following [?], we add two fully connected layers before the classifier. The two layers first map the 512-dimensional features to a 3-dimensional feature sphere and then map back to the 10-dimensional feature space for prediction. The contrastive loss $\mathcal{L}_{con}$ is enforced on the 3-dimensional features. After training, we visualize the 3-dimensional features learned by ResNet-18 in MATLAB.

## C.2  More Details of Image Classification

**Dataset details.** Following [27], we test on 9 natural image datasets, including ImageNet20 (a subset of ImageNet with 20 classes) [11], CIFAR10, CIFAR100 [29], Caltech-101 [15], DTD [10], FGVC Aircraft [39], Standard Cars [28], Oxford-IIIT Pets [44] and Oxford 102 Flowers [42]. In addition, considering real-world datasets may be class-imbalanced [67, 68? ? , 70], we also evaluate Core-tuning on the iNaturalist18 dataset [50]. Most datasets are obtained from their official websites, except ImageNet20 and Oxford 102 Flowers. The ImageNet20 dataset is obtained by combining two open-source ImageNet subsets with 10 classes, *i.e.,* ImaegNette and ImageWoof [21]. Moreover, Oxford 102 Flowers is obtained from Kaggle[2]. These datasets cover a wide range of classification tasks, including coarse-grained object classification (*i.e.,* ImageNet20, CIFAR, Caltech-101), fine-grained object classification (*i.e.,* Cars, Aircraft, Pets) and texture classification (*i.e.,* DTD). The statistics of all datasets are reported in Table 1.

Table 1: Statistics of datasets.

| DataSet | #Classes | # Training | # Test |
|---|---|---|---|
| ImageNet20 [21, 11] | 20 | 18,494 | 7,854 |
| CIFAR10 [29] | 10 | 50,000 | 10,000 |
| CIFAR100 [29] | 100 | 50,000 | 10,000 |
| Caltech-101 [15] | 102 | 3,060 | 6,084 |
| Describable Textures (DTD) [10] | 47 | 3,760 | 1,880 |
| FGVG Aircraft [39] | 100 | 6,667 | 3,333 |
| Standard Cars [28] | 196 | 8,144 | 8,041 |
| Oxford-IIIT Pets [44] | 37 | 3,680 | 3,369 |
| Oxford 102 Flowers [42] | 102 | 6,552 | 818 |
| iNaturalist18 [50] | 8,142 | 437,513 | 24,426 |

**Implementation details.** We implement all methods in PyTorch. All checkpoints of self-supervised models are provided by the authors or by the PyContrast GitHub repository[3]. For most datasets, following [6, 27], we preprocess images via random resized crops to $224\times224$ and flips. At the test time, we resize images to $256\times256$ and then take a $224\times224$ center crop. In such a setting, however, we find it difficult to reproduce the performance of some CSL models [6]. Therefore, for some datasets (*e.g.,* CIFAR10 and Aircraft), we resize images to different scales and use rotation augmentations. Although the preprocessing of some datasets is slightly different from [6], the results in this paper are obtained with the same preprocessing method *w.r.t.* each dataset and thus are fair.

Following [27], we initialize networks with the checkpoints of contrastive self-supervised models. For most datasets, we fine-tune networks for 100 epochs using Nesterov momentum via the cosine learning rate schedule. For ImageNet20, we fine-tune networks using stochastic gradient descent via the linear learning rate decay. For iNaturalist18, we fine-tune networks for 160 epochs. For all datasets, the momentum parameter is set to 0.9, while the factor of weight decay is set to $10^{-4}$. As for Core-tuning, we set the clipping thresholds of hard negative generation to be $\lambda_n$=0.8 and the temperature $\tau$=0.07. The dimension of the contrastive features is 256 and the depth of non-linear projection is 2 layers. Following [6], we perform hyper-parameter tuning for each dataset. Specifically, we select the batch size from $\{64, 128, 256\}$, the initial learning rate from $\{0.01, 0.1\}$ and $\eta/\alpha$ from $\{0.1, 1, 10\}$. The experiments are conducted on 4 TITAN RTX 2080 GPUs for iNaturalist18, and 1 GPU for all other datasets. All results are averaged over 3 runs. We adopt the top-1 accuracy as the metric. The statistics of the used hyper-parameters are provided in Table 2. For other baselines, we use the same training setting for each dataset, and tune their hyper-parameters as best as possible.

---

[2]`https://www.kaggle.com/c/oxford-102-flower-pytorch`.
[3]https://github.com/HobbitLong/PyContrast

Table 2: Statistics of the used hyper-parameters in Core-tuning.

| Hyper-parameter | ImageNet20 | CIFAR10 | CIFAR100 | Caltech101 | DTD | Aircraft | Cars | Pets | Flowers | iNarutalist18 |
|---|---|---|---|---|---|---|---|---|---|---|
| epochs | | | | | 100 | | | | | 160 |
| batch size | 256 | 256 | 256 | 256 | 256 | 64 | 64 | 64 | 64 | 128 |
| loss trade-off factor $\eta$ | 0.1 | 0.1 | 1 | 1 | 0.1 | 0.1 | 0.1 | 0.1 | 1 | 10 |
| mixup factor $\alpha$ | 1 | 1 | 0.1 | 0.1 | 1 | 0.1 | 0.1 | 1 | 0.1 | 1 |
| learning rate (lr) | 0.1 | 0.01 | 0.01 | 0.01 | 0.01 | 0.01 | 0.01 | 0.01 | 0.01 | 0.1 |
| lr schedule | linear | | | | cosine decay | | | | | |
| temperature $\tau$ | | | | | 0.07 | | | | | |
| threshold $\lambda_n$ | | | | | 0.8 | | | | | |
| weight decay factor | | | | | $10^{-4}$ | | | | | |
| momentum factor | | | | | 0.9 | | | | | |
| projection dimension | | | | | 256 | | | | | |
| projection depth | | | | | 2 layers | | | | | |

## C.3 More Details of Domain Generalization

**Dataset details.** We use 3 benchmark datasets, *i.e.,* PACS [32], VLCS [14] and Office-Home [51]. The data statistics are shown in Table 3, where each dataset has 4 domains. In each setting, we select 3 domains to fine-tune the networks and then test on the rest of the unseen domains. The key challenge is the distribution discrepancies among domains, leading to poor performance of neural networks on the target domain [? ? ? ? ? ].

Table 3: Statistics of datasets.

| DataSet | #Domains | #Classes | #Samples | Size of images |
|---|---|---|---|---|
| PACS | 4 | 7 | 9,991 | (3,224,224) |
| VLCS | 4 | 5 | 10,729 | (3,224,224) |
| Office-Home | 4 | 65 | 15,588 | (3,224,224) |

**Implementation details.** The overall scheme of Core-tuning for domain generalization is shown in Figure 1. The experiments are implemented based on the DomainBed repository [? ] in PyTorch. During fine-tuning, we preprocess images through random resized crops to $224 \times 224$, horizon flips, color jitter and random gray scale. At the test time, we directly resize images to $224 \times 224$. We initialize ResNet-50 with the weights of the MoCo-v2 pre-trained model, and fine-tune it for 20,000 steps at a batch size of 32 using the Adam optimizer on a single TITAN RTX 2080 GPU. We set the initial learning rate to $5 \times 10^{-5}$ and adjust it via the exponential learning rate decay. All other hyper-parameters of Core-tuning are the same as image classification. Besides, we use Accuracy as the metric in domain generalization.

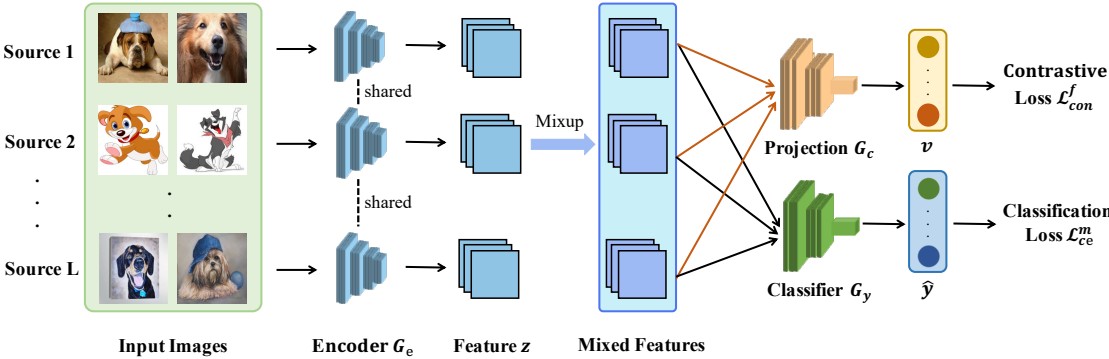

Figure 1: The overall scheme of Core-tuning in the setting of cross-domain generalization.

## C.4 Implementation Details of Robustness Training

We conduct this experiment in PyTorch. We take Caltech-101, DTD, Pets, and CIFAR10 as datasets, whose preprocessing are the same as the ones in image classification. We use MoCo-v2 pre-trained ResNet-50 as the backbone, and use Projected Gradient Descent (PGD) [38] to generate adversarial samples. During adversarial training (AT), we use both clean and adversarial samples for training with various fine-tuning methods on a single TITAN RTX 2080 GPU. Other training schemes (*e.g.,* the optimizer, the hyper-parameters, the learning rate scheme) are the same as image classification.

# D More Experimental Results

## D.1 More Results on Domain Generalization

This appendix further reports the results of domain generalization on OfficeHome. The observations from Table 4 are same to the main text. First, when fine-tuning with cross-entropy, the contrastive self-supervised model performs worse than the supervised pre-trained model. This results from the relatively worse discriminative abilities of the contrastive self-supervised model, which can also be found in Table 1 of the main paper. Second, enforcing contrastive regularizer during fine-tuning improves domain generalization performance, since the contrastive regularizer helps to learn more discriminative features (cf. Theorem 1) and also helps to alleviate distribution shifts among domains [24], hence leading to better performance. Last, Core-tuning further improves the generalization performance of models on all datasets. This is because hard pair generation further boosts contrastive learning, while smooth classifier learning also benefits model generalizability. We thus conclude that Core-tuning improves model generalization on downstream tasks.

Table 4: Domain generalization accuracies of various fine-tuning methods for MoCo-v2 pre-trained ResNet-50 the on Office-Home dataset. CE means cross-entropy; CE-Con enhances CE with the contrastive loss. Here, A/C/P/R are four domains in Office-Home.

| Pre-training | Fine-tuning | Office-Home | | | | |
|---|---|---|---|---|---|---|
| | | A | C | P | R | Avg. |
| Supervised | CE | 56.08 | 50.83 | 72.49 | 75.21 | 63.82 |
| MoCo-v2 | CE | 50.31 | 48.91 | 64.72 | 68.76 | 58.18 |
| | CE-Con | 55.87 | 50.23 | 71.51 | 74.99 | 63.15 |
| | ours | **58.70** | **52.43** | **72.89** | **75.36** | **64.85** |

## D.2 More Results on Adversarial Training

In the main paper, we apply Core-tuning to adversarial training on CIFAR10, while this appendix further provides the results of adversarial training on three other natural image datasets, *i.e.,* Caltech-101, DTD and Pets. We draw several observations based on the results on 3 image datasets in Table 5. First, despite good clean accuracy, standard fine-tuning with cross-entropy cannot defend against adversarial attack, leading to poor robust accuracy. Second, AT with cross-entropy improves the robust accuracy significantly, but it inevitably degrades the clean accuracy due to the accuracy-robustness trade-off [49]. In contrast, the contrastive regularizer improves both robust and clean accuracies. This is because contrastive learning helps to improve robustness generalization (*i.e.,* alleviating the distribution shifts between clean samples and adversarial samples), thus leading to better performance. Last, Core-tuning further boosts AT and, surprisingly, even achieves better clean accuracy than the standard fine-tuning under the $\ell_2$ attack. To our knowledge, this is quite promising since even the most advanced AT methods [61, 65] find it difficult to conquer the accuracy-robustness trade-off [63]. The improvement is mainly derived from that both contrastive learning and smooth classifier learning boost the robustness generalization. We thus conclude that Core-tuning is beneficial to model robustness. We also hope that Core-tuning can motivate people to rethink the accuracy-robustness trade-off in adversarial training in the future.

Table 5: Adversarial training performance of MoCo-v2 pre-trained ResNet-50 under the attack of PGD-10 in terms of robust and clean accuracies. CE indicates cross-entropy; AT-CE indicates adversarial training (AT) with CE; AT-CE-Con enhances AT-CE with the contrastive loss; AT-ours uses Core-tuning for AT.

| Method | PGD - $\ell_2$ attack ($\epsilon = 0.5$) | | | | | | PGD - $\ell_\infty$ attack ($\epsilon = 4/255$) | | | | | |
|---|---|---|---|---|---|---|---|---|---|---|---|---|
| | Caltech101 | | DTD | | Pets | | Caltech101 | | DTD | | Pets | |
| | Robust | Clean | Robust | Clean | Robust | Clean | Robust | Clean | Robust | Clean | Robust | Clean |
| CE | 55.69 | 91.87 | 42.25 | 71.68 | 30.94 | 89.05 | 27.03 | **91.87** | 18.37 | **71.68** | 4.63 | **89.05** |
| AT-CE | 87.35 | 91.61 | 61.93 | 68.81 | 78.67 | 86.25 | 78.61 | 90.65 | 47.27 | 67.13 | 63.59 | 84.21 |
| AT-CE-Con | 88.67 | 92.61 | 64.75 | 71.24 | 79.53 | 87.01 | 79.87 | 91.08 | 48.95 | 69.07 | 65.60 | 86.85 |
| AT-ours | **89.21** | **92.83** | **66.49** | **72.94** | **82.54** | **89.22** | **80.73** | 91.64 | **49.43** | 70.65 | **67.98** | 87.20 |

### D.3 More Results on Image Classification

**The results with standard errors.** In the main paper, we report the results of image classification and ablations studies on 9 natural image datasets in terms of the average accuracy. To make the results more complete, this appendix further reports the results with their standard errors (cf. Tables 6-7).

Table 6: Comparisons of various fine-tuning methods for MoCo-v2 pre-trained ResNet-50 on image classification in terms of top-1 accuracy. Here, "Avg." indicates the average accuracy over 9 datasets. SL-CE-tuning denotes supervised pre-training on ImageNet and then fine-tuning with cross-entropy.

| Algorithm | ImageNet20 | CIFAR10 | CIFAR100 | Caltech101 | DTD |
|---|---|---|---|---|---|
| SL-CE-tuning | 91.01+/-1.27 | 94.23+/-0.07 | 83.40+/-0.12 | 93.65+/-0.21 | 74.40+/-0.45 |
| CE-tuning | 88.28+/-0.47 | 94.70+/-0.39 | 80.27+/-0.60 | 91.87+/-0.18 | 71.68+/-0.53 |
| L2SP [35] | 88.49+/-0.40 | 95.14+/-0.22 | 81.43+/-0.22 | 91.98+/-0.07 | 72.18+/-0.61 |
| M&M [62] | 88.53+/-0.21 | 95.02+/-0.07 | 80.58+/-0.19 | 92.91+/-0.08 | 72.43+/-0.43 |
| DELTA [33] | 88.35+/-0.41 | 94.76+/-0.05 | 80.39+/-0.41 | 92.19+/-0.45 | 72.23+/-0.23 |
| BSS [9] | 88.34+/-0.62 | 94.84+/-0.21 | 80.40+/-0.30 | 91.95+/-0.12 | 72.22+/-0.17 |
| RIFLE [34] | 89.06+/-0.28 | 94.71+/-0.13 | 80.36+/-0.07 | 91.94+/-0.23 | 72.45+/-0.30 |
| SCL [18] | 89.29+/-0.07 | 95.33+/-0.09 | 81.49+/-0.27 | 92.84+/-0.03 | 72.73+/-0.31 |
| Bi-tuning [71] | 89.06+/-0.08 | 95.12+/-0.15 | 81.42+/-0.01 | 92.83+/-0.06 | 73.53+/-0.37 |
| Core-tuning | **92.73+/-0.17** | **97.31+/-0.10** | **84.13+/-0.27** | **93.46+/-0.06** | **75.37+/-0.37** |

| Algorithm | Aircraft | Cars | Pets | Flowers | Avg. |
|---|---|---|---|---|---|
| SL-CE-tuning | 87.03+/-0.02 | 89.77+/-0.11 | 92.17+/-0.12 | 98.78+/-0.10 | 89.35 |
| CE-tuning | 86.87+/-0.18 | 88.61+/-0.43 | 89.05+/-0.01 | 98.49+/-0.06 | 87.76 |
| L2SP [35] | 86.55+/-0.30 | 89.00+/-0.23 | 89.43+/-0.27 | 98.66+/-0.20 | 88.10 |
| M&M [62] | 87.45+/-0.28 | 88.90+/-0.70 | 89.60+/-0.09 | 98.57+/-0.15 | 88.22 |
| DELTA [33] | 87.05+/-0.37 | 88.73+/-0.05 | 89.54+/-0.48 | 98.65+/-0.17 | 87.99 |
| BSS [9] | 87.18+/-0.71 | 88.50+/-0.02 | 89.50+/-0.42 | 98.57+/-0.15 | 87.94 |
| RIFLE [34] | 87.60+/-0.50 | 89.72+/-0.11 | 90.05+/-0.26 | 98.70+/-0.06 | 88.29 |
| SCL [18] | 87.44+/-0.31 | 89.37+/-0.13 | 89.71+/-0.20 | 98.65+/-0.10 | 88.54 |
| Bi-tuning [71] | 87.39+/-0.01 | 89.41+/-0.28 | 89.90+/-0.06 | 98.57+/-0.10 | 88.58 |
| Core-tuning | **89.48+/-0.17** | **90.17+/-0.03** | **92.36+/-0.14** | **99.18+/-0.15** | **90.47** |

Table 7: Ablation studies of Core-tuning (Row 5) for fine-tuning MoCo-v2 pre-trained ResNet-50 on 9 natural image datasets in terms of top-1 accuracy. Here, "Avg." indicates the average accuracy over the 9 datasets. Besides, $\mathcal{L}_{con}$ is the original supervised contrastive loss, while $\mathcal{L}_{con}^{f}$ is our focal contrastive loss. Moreover, "mix" denotes the manifold mix, while "mix-H" indicates the proposed hardness-directed mixup strategy in our method.

| $\mathcal{L}_{ce}$ | $\mathcal{L}_{con}$ | $\mathcal{L}_{con}^{f}$ | mix | mix-H | ImageNet20 | CIFAR10 | CIFAR100 | Caltech101 | DTD |
|---|---|---|---|---|---|---|---|---|---|
| √ | | | | | 88.28+/-0.47 | 94.70+/-0.39 | 80.27+/-0.60 | 91.87+/-0.18 | 71.68+/-0.53 |
| √ | √ | | | | 89.29+/-0.07 | 95.33+/-0.09 | 81.49+/-0.27 | 92.84+/-0.03 | 72.73+/-0.31 |
| √ | | √ | | | 90.67+/-0.09 | 95.43+/-0.06 | 81.03+/-0.11 | 92.68+/-0.06 | 73.31+/-0.40 |
| √ | √ | | | √ | 92.20+/-0.15 | 97.01+/-0.10 | 83.89+/-0.20 | 93.22+/-0.18 | 74.78+/-0.31 |
| √ | | √ | | √ | **92.73+/-0.17** | **97.31+/-0.10** | **84.13+/-0.27** | **93.46+/-0.06** | **75.37+/-0.37** |

| $\mathcal{L}_{ce}$ | $\mathcal{L}_{con}$ | $\mathcal{L}_{con}^{f}$ | mix | mix-H | Aircraft | Cars | Pets | Flowers | Avg. |
|---|---|---|---|---|---|---|---|---|---|
| √ | | | | | 86.87+/-0.18 | 88.61+/-0.43 | 89.05+/-0.01 | 98.49+/-0.06 | 87.76 |
| √ | √ | | | | 87.44+/-0.31 | 89.37+/-0.13 | 89.71+/-0.20 | 98.65+/-0.10 | 88.54 |
| √ | | √ | | | 88.37+/-0.14 | 89.06+/-0.14 | 91.37+/-0.03 | 98.74+/-0.12 | 88.96 |
| √ | √ | | | √ | 88.88+/-0.34 | 89.79+/-0.12 | 91.95+/-0.33 | 98.94+/-0.12 | 90.07 |
| √ | | √ | | √ | **89.48+/-0.17** | **90.17+/-0.03** | **92.36+/-0.14** | **99.18+/-0.15** | **90.47** |

**The fine-tuning results on ImageNet.** Since ImageNet has rich labeled samples for fine-tuning and the CSL models are also pre-trained on ImageNet, the performance gain of different fine-tuning methods may not vary as significantly as on the small-scale target datasets. Even so, the results in Table 8 also demonstrate the effectiveness of Core-tuning on very large-scale data.

Table 8: Fine-tuning results of the MoCo-v2 ResNet-50 fine-tuned by various methods, on ImageNet.

| Pre-training | Fine-tuning | Top-1 accuracy |
|---|---|---|
| MoCo-v2 [8] | CE-tuning | 76.82 |
| MoCo-v2 [8] | CE-Contrastive-tuning | 77.13 |
| MoCo-v2 [8] | Core-tuning (ours) | **77.43** |

**More results on different pre-training methods.** This appendix provides the fine-tuning results of Core-tuning for the SimCLR pre-trained models. Since the official checkpoints of SimCLR-v1 [6] and SimCLR-v2 [7] are based on Tensorflow, we convert them to the PyTorch and try to reproduce cross-entropy tuning (CE-tuning) in our experimental settings. Note that although the reproduction performance of CE-tuning is slightly worse than the original paper [6, 7], the results in this paper are obtained with the same preprocessing method *w.r.t.* each dataset and thus are fair. As shown in Table 9, Core-tuning consistently outperforms CE-tuning for SimCLR pre-trained models.

Table 9: Fine-tuning results of ResNet-50, pre-trained by various methods.

| Pre-training | Caltech101 | | DTD | | Pets | |
|---|---|---|---|---|---|---|
| | CE-tuning | ours | CE-tuning | ours | CE-tuning | ours |
| SimCLR-v1 [6] | 90.53+/-0.06 | **92.40+/-0.06** | 90.53+/-0.06 | **71.26+/-0.05** | 89.34+/-0.46 | **90.89+/-0.09** |
| SimCLR-v2 [7] | 92.44+/-0.18 | **93.46+/-0.02** | 71.26+/-0.26 | **74.75+/-0.41** | 88.28+/-0.26 | **90.64+/-0.31** |

**The results on linear evaluation.** This appendix provides linear evaluation for Core-tuning. Specifically, we first fine-tune the MoCo-v2 pre-trained ResNet-50 with Core-tuning and then train a linear classifier for prediction. As shown in Table 10, Core-tuning performs better than CE-tuning.

Table 10: Results of linear evaluation for the ResNet-50 fine-tuned by various methods, on CIFAR10.

| Pre-training | Fine-tuning | Top-1 accuracy |
|---|---|---|
| MoCo-v2 [8] | CE-tuning | 94.78+/-0.28 |
| MoCo-v2 [8] | Core-tuning (ours) | **97.09+/-0.14** |

**The results on KNN evaluation.** This appendix provides the KNN evaluation for Core-tuning. To be specific, we first fine-tune the MoCo-v2 pre-trained ResNet-50 with Core-tuning and then use KNN for prediction. As shown in Table 11, Core-tuning also outperforms CE-tuning.

Table 11: Results of KNN evaluation for the ResNet-50 fine-tuned by various methods, on CIFAR10.

| Pre-training | Fine-tuning | Top-1 accuracy |
|---|---|---|
| MoCo-v2 [8] | CE-tuning | 94.63+/-0.32 |
| MoCo-v2 [8] | Core-tuning (ours) | **96.65+/-0.06** |

## D.4 The Results with Standard Errors on Semantic Segmentation

In the main paper, we report the average results of semantic segmentation on PASCAL VOC. This appendix further reports the results with their standard errors (cf. Table 12).

Table 12: Fine-tuning performance on PASCAL VOC semantic segmentation based on DeepLab-V3 with ResNet-50, pre-trained by various CSL methods. CE indicates cross-entropy.

| Pre-training | Fine-tuning | MPA | FWIoU | MIoU |
|---|---|---|---|---|
| Supervised | CE | 87.10+/-0.20 | 89.12+/-0.17 | 76.52+/-0.34 |
| InsDis [58] | CE | 83.64+/-0.12 | 88.23+/-0.08 | 74.14+/-0.21 |
| | ours | **84.53+/-0.31** | **88.67+/-0.07** | **74.81+/-0.13** |
| PIRL [41] | CE | 83.16+/-0.26 | 88.22+/-0.24 | 73.99+/-0.42 |
| | ours | **85.30+/-0.24** | **88.95+/-0.08** | **75.49+/-0.36** |
| MoCo-v1 [20] | CE | 84.71+/-0.56 | 88.75+/-0.04 | 74.94+/-0.12 |
| | ours | **85.70+/-0.32** | **89.19+/-0.02** | **75.94+/-0.23** |
| MoCo-v2 [8] | CE | 87.31+/-0.31 | 90.26+/-0.12 | 78.42+/-0.28 |
| | ours | **88.76+/-0.34** | **90.75+/-0.04** | **79.62+/-0.12** |
| SimCLR-v2 [7] | CE | 87.37+/-0.48 | 90.27+/-0.12 | 78.16+/-0.19 |
| | ours | **87.95+/-0.20** | **90.71+/-0.13** | **79.15+/-0.33** |
| InfoMin [47] | CE | 87.17+/-0.20 | 89.84+/-0.09 | 77.84+/-0.24 |
| | ours | **88.92+/-0.36** | **90.65+/-0.09** | **79.48+/-0.30** |

# E More Analysis of Core-tuning

## E.1 Analysis of Projection Dimension and Depth

In previous experiments, we use a 2-layer MLP to extract contrastive features with dimension 256. Here, we further analyze how the dimension and the depth influence Core-tuning. The results on ImageNet20 are reported in Figure 2, where the fine-tuning performance of Core-tuning can be further improved by changing the feature dimension to 128 and the depth to 3. Note that the best dimension and depth of the projection head may vary on different datasets, but the default setting (*i.e.,* dimension 256 and depth 2) is enough to obtain consistently good performance.

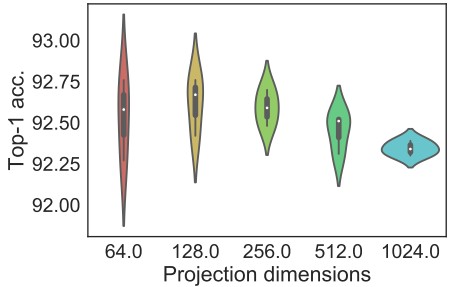 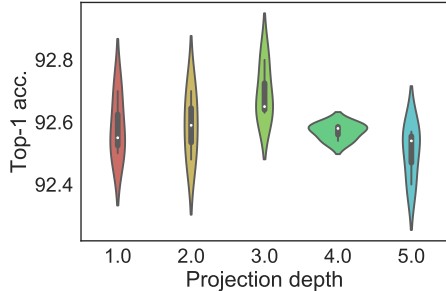

Figure 2: Analysis of the projection dimension and the projection depth in Core-tuning on ImageNet20 based on MoCo-v2 pre-trained ResNet-50. Each run tests one parameter and fixes others. Best viewed in color.

## E.2 Analysis of Loss and Mixup Hyper-Parameters

This appendix discusses the influence of the loss trade-off parameter $\eta$ and the mixup sampling factor $\alpha$ on Core-tuning based on the ImageNet20 dataset. Each run tests one parameter and fixes others. As shown in Figure 3, when $\eta$=0.1 and $\alpha$=1, Core-tuning performs slightly better on ImageNet20. Note that the best $\eta$ and $\alpha$ can be different on diverse datasets.

## E.3 Analysis of Temperature Factor

Following the implementation of the supervised contrastive loss [25], we set the temperature factor $\tau$ to 0.07 for Core-tuning by default. In this section, we further analyze the influence of $\tau$ on Core-tuning when fine-tuning MoCo-v2 pre-trained models on ImageNet20. As shown in Figure 3, when $\tau$ is small (*e.g.,* 0.01 or 0.07), Core-tuning performs slightly better on ImageNet20. The potential reason is that a small temperature parameter implicitly helps the method to learn hard positive/negative pairs [**?** ], which are more informative and beneficial to contrastive learning. Note that the best $\tau$ can be different on different datasets, but the default setting (*i.e.,* $\tau = 0.07$) is enough to achieve comparable performance.

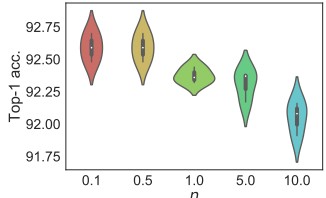 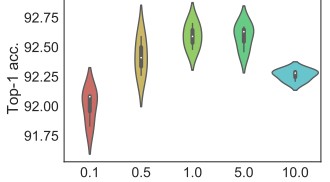 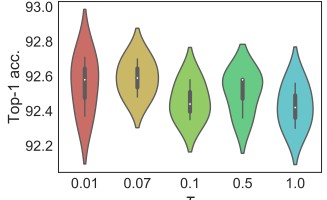

Figure 3: Analysis of $\eta$, $\alpha$ and the temperature factor in Core-tuning on ImageNet20 based on MoCo-v2 pre-trained ResNet-50. Each run tests one factor and fixes others. Best viewed in color.

### E.4 Analysis of Hard Pair Thresholds

In our hardness-directed mixup strategy, to make the generated negative pairs closer to negative pairs, we clip $\lambda \sim \text{Beta}(\alpha, \alpha)$ by $\lambda \geq \lambda_n$ when generating hard negative pairs. In our experiments, we set the threshold $\lambda_n = 0.8$. In this appendix, we analyze the influences of the negative pair threshold $\lambda_n$. Meanwhile, although we do not constrain hard positive generation, we also analyze the potential positive pair threshold $\lambda_p$. The results on ImageNet20 are reported in Table 3. On the one hand, $\lambda_n$ satisfies our expectation that the generated hard negative pairs should be closer to negatives, *i.e.,* a larger $\lambda_n$ can lead to better performance. On the other hand, we find when no crop is conducted for hard positive generation (*i.e.,* $\lambda_p=0$), the performance is slightly better. We conjecture that since the generated hard positives are located in the borderline area between positives and negatives, allowing the generated hard positives to close to negatives may have a margin effect on contrastive learning and thus boosts performance. Despite this, Core-tuning with a large $\lambda_p$ performs similarly well.

Table 13: Threshold analysis for hard pair generation in Core-tuning on ImageNet20 based on MoCo-v2 pre-trained ResNet-50. Each run tests one parameter and fixes another one to 0.8.

| Thresholds | 0 | 0.2 | 0.4 | 0.6 | 0.8 |
|---|---|---|---|---|---|
| Negative pair threshold $\lambda_n$ | 91.55 | 91.94 | 92.19 | 92.36 | 92.59 |
| Positive pair threshold $\lambda_p$ | 92.73 | 92.68 | 92.64 | 92.60 | 92.59 |

### E.5 Relationship Between Pre-Training and Fine-Tuning Accuracies

We further explore the relationship between ImageNet performance and Core-tuning fine-tuning performance on Caltech-101 for various contrastive self-supervised models. Here, the ImageNet performance of a contrastive self-supervised model is obtained by training a new linear classifier on the frozen pre-trained representation and then evaluate the model on the ImageNet test set. For convenience, we directly follow the ImageNet performance reported in the original paper of the corresponding methods. As shown in Figure 4, the fine-tuning result of each contrastive self-supervised model on Caltech-101 is highly correlated with the model result on ImageNet. This implies that the ImageNet performance can be a good predictor for the fine-tuning performance of contrastive self-supervised models. Such a finding is consistent with supervised pre-trained models [27]. Even so, note that the correlation is not perfect, where a contrastive pre-trained model with better ImageNet performance does not necessarily mean better fine-tuning performance, *e.g.,* SimCLR-v2 vs MoCo-v2.

### E.6 Effectiveness of Hard Pair Generation for Contrastive Fine-Tuning

In our proposed Core-tuning, we use all the generated positive sample pairs and the original samples as positive pairs for contrastive fine-tuning. In this appendix, to better evaluate the effectiveness of hard pair generation, we do not use original data as positive pairs but only use the generated hard positive pairs for contrastive learning. As shown in Table 14, only using the generated hard positive pairs for contrastive learning is enough to obtain comparable performance. Such results further verify the effectiveness of our hardness-directed mixup strategy as well as the importance of hard positive pairs for contrastive fine-tuning.

Table 14: Comparisons with only using the generating hard positive pairs for contrast on CIFAR10.

| Pre-training | Fine-tuning | The used positive pairs for contrast? | Top-1 accuracy |
|---|---|---|---|
| MoCo-v2 [8] | CE-tuning | × | 94.70+/-0.39 |
| MoCo-v2 [8] | Core-tuning | only the generated hard positive pairs | 97.31+/-0.09 |
| MoCo-v2 [8] | Core-tuning | all positive pairs | 97.31+/-0.10 |

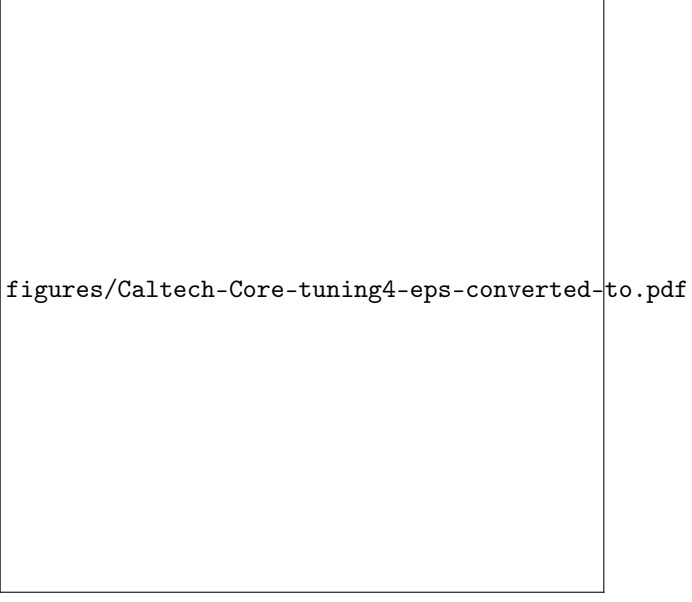

Figure 4: The relationship between ImageNet performance and Core-tuning fine-tuning performance on Caltech-101 for contrastive self-supervised ResNet-50 models. Better viewed in color.

### E.7  Effectiveness of Smooth Classifier Learning

In Core-tuning, to better exploit the learned discriminative feature space by contrastive fine-tuning, we use the mixed samples for classifier training, so that the classifier can be more smooth and far away from the original training data. In this appendix, to better evaluate the effectiveness of smooth classifier learning, we compare Core-tuning with a variant that does not use the mixed data for classifier learning. As shown in Table 15, smooth classifier learning contributes to the fine-tuning performance of contrastive self-supervised models on downstream tasks. The results demonstrate the effectiveness of smooth classifier learning and also show its importance in Core-tuning.

Table 15: Influence of smooth classifier learning on CIFAR10.

| Pre-training | Fine-tuning | Smooth classifier learning? | Top-1 accuracy |
|---|---|---|---|
| MoCo-v2 [8] | CE-tuning | $\times$ | 94.70+/-0.39 |
| MoCo-v2 [8] | CE-tuning | $\sqrt{}$ | 95.43+/-0.20 |
| MoCo-v2 [8] | Core-tuning | $\times$ | 96.13+/-0.11 |
| MoCo-v2 [8] | Core-tuning | $\sqrt{}$ | 97.31+/-0.10 |