# OpenReview forum: "Unleashing the Power of Contrastive Self-Supervised Visual Models via Contrast-Regularized Fine-Tuning"
_NeurIPS.cc/2021/Conference — NeurIPS 2021 Poster_

### Official Review · Reviewer_KA1Y · 2021-06-29

**Rating:** 8
**Confidence:** 5

**Summary:**

This paper proposed a new problem: how to fine-tune a contrastive pretrained model, and proposed hard example minining and smooth boundary to help contrastive fine-tuning. The proposed Core-tuning method has been theoretically and empirically validated on a variety of tasks.

**Limitations And Societal Impact:**

There are discussions of main limitations. The BYOL type is a good plus.

**Main Review:**

Overall, I like this paper, which is well-written and the reading is quite smooth.

On the positive side, the theoretical bounds are both reasonable and give insights to the problem. The proposed methods have also brought about significant accuracy improvements. Ablation studies are extensive, and its application to semantic segmentation, robustness, and domain adaptation are also interesting.

Table 1 shows that supervised pretrained model indeed fine-tunes better than self-supervised models. Some earlier self-supervised learning methods have used misleading setups to lead to a false claim. It is useful that the author(s) pointed this out in Table 1 and Line 243.

But I also have some (relatively) minor issues to be clarified by the author(s).

First, Core-tuning is designed for contrastive methods. However, if we ignore the theoretical side, details of the algorithm did not depend on the InfoNCE loss. Hence, I wonder if Core-tuning is applied to BYOL, will it still help? If Core-tuning works for BYOL and other non-contrastive self-supervised methods, it will be more useufl.

Second, in Line 115 and the pseudo code (in appendix), it is not clear whether the features are extracted by Ge and fixed? In other words, will the feature learning part (backbone network) be updated in the fine-tuning?

Third, in Line 181, what is an anchor? It is not defined.

Fourth, the difference between the last 2 rows in Table 2 is small (but they are indeed consistent). So, some ablation studies are missing. What if we use supervised pretrained model + mixup + CE as a baseline (standard mixup, not manifold)? Our experience tells us that mixup + CE is signfiicantly better than CE alone in fine-tuning (both supervised and self-supervised) pretrained models.

Fifth, Table 4 is not that diverse -- they are mostly 1 architecture. In fact, it is more interesting to see results on different models like MobileNet (compact), EfficientNet (accuate), etc.

**Time Spent Reviewing:**

1.2

---

> ### Author Response · Authors · 2021-08-10
> **Response**
>
> Thanks very much for your constructive comments on our paper, particularly for recognizing the value of our theoretical analysis. We expect that our theoretical result on contrastive loss can motivate more work in the future. We next address the main concerns as follows.
> ***
> **Q1. More results on BYOL pre-trained models.**
>
> A1. As shown in the following table, Core-tuning also fine-tunes BYOL pre-trained ResNet-50 well. Since Table 3 (in the submission) has shown that core-tuning fine-tunes clustering self-supervised models well too, our proposed method is generalizable and can well fine-tune the models pre-trained by non-contrastive methods.
>
>
> |  Model   | Pre-training  | Fine-tuning  | Caltech101  | DTD  | Pets |
> |  ----  | :----: | ---- | :----: | :----: | :----: |
> |  ResNet-50  |     BYOL  |       CE-tuning            |        91.19   |   	74.94  |  	92.39 |
> |  ResNet-50  |     BYOL   |       Core-tuning  |        93.25     | 	76.56   | 	93.74 |
>
> ***
> **Q2. In Line 115 and the pseudo-code (in appendix), it is not clear whether the features are extracted by $G_e$ and fixed?**
>
> A2. The feature extractor is updated during the fine-tuning, and it is the key part that the proposed contrastive regularizer improves.
>
> ***
> **Q3. In Line 181, what is an anchor?**
>
> A3. As shown in Line 125-127, the concept of “anchor”, positive and negative sample pairs are defined to compute the contrastive loss in Eqn. (1). To be specific, the contrastive loss pulls the anchor and positive pairs closer and pushes the anchor and negative pairs apart. After that and in Line 181, we follow this definition.
>
> ***
> **Q4. The baseline of supervised pretrained model + image mixup + CE.**
>
> A4. In fact, we have shown the beneficial effect of mixup (manifold mixup) on self-supervised model fine-tuning in Table 2, i.e., Row 3 (CE+manifold mixup) v.s. Row 1 (CE). Here, following the suggestion, we further use CE+standard mixup to fine-tune supervised pre-trained models. As shown in the following table,  standard mixup also improves the fine-tuning performance, but  the improvement is not as significant as our method.
>
> |  Model   | Pre-training  | Fine-tuning  | Caltech101  | DTD  | Pets |
> |  ----  | :----: | ---- | :----: | :----: | :----: |
> |  ResNet-50  |     supervised  |    CE-tuning     |       93.65      |  	74.40    |  	92.17 |
> |  ResNet-50  |     supervised  |    CE+data mixup      |        93.76     |   	74.85 |  93.34 |
> |  ResNet-50  |     supervised   |       Core-tuning  |        94.20   |     	77.27   |   	93.82 |
>
> ***
> **Q5. It is interesting to see results on more diverse network architectures, like MobileNet.**
>
> A5. In Table 4, we conduct experiments based on ResNet-type architectures, since almost all contrastive self-supervised learning (CSL) studies only provide ResNet-type pre-trained weights. Due to the time limit, we did not finish CSL training for MobileNets and we also did not find any publically available pre-trained weights either. Therefore, we provide an empirical study on supervised pre-trained MobileNet-V2. The results in the following table show that the proposed Core-tuning also works well on MobileNet-type architectures.
>
> |  Model   | Pre-training  | Fine-tuning  | Caltech101  | DTD  | Pets |
> |  ----  | :----: | ---- | :----: | :----: | :----: |
> |  MobileNet-V2 |     supervised  |       CE-tuning       |        88.96      |  	68.07    |  87.98  |
> |  MobileNet-V2 |     supervised   |    Core-tuning  |        92.07     |   	71.77    |  92.77  |
>
> In addition to convolutional networks, we also try to use Core-tuning to fine-tune the vision transformer model (DeiT[1]), pre-trained by a non-contrastive self-supervised method (DINO [2]). The results in the below table further demonstrate that our method is effective and performs better than CE-tuning for fine-tuning self-supervised vision transformers. We will add the results in the revision.
>
> |  Model   | Pre-training  | Fine-tuning  | Caltech101  | DTD  | Pets |
> |  ----  | :----: | ---- | :----: | :----: | :----: |
> |  DeiT-S/16 [1]  |     DINO [2]  |       CE-tuning       |        91.24      |        71.35      |     92.43  |
> |  DeiT-S/16 [1]  |     DINO [2]    |    Core-tuning  |       92.31         |     72.83      |    93.72 |
>
> &nbsp;
>
> [1] Training data-efficient image transformers & distillation through attention, arxiv, 2021.
>
> [2] Emerging Properties in Self-Supervised Vision Transformers, arxiv, 2021.

---

### Official Review · Reviewer_1Skd · 2021-07-13

**Rating:** 6
**Confidence:** 5

**Summary:**

This paper proposes a fine-tuning method called core-tuning for Contrastive Self-supervised Learning (CSL). Core-tuning employs contrastive loss as a regularizer for preserving intra-class features and further improves the finetuned performance. Besides, hard sample mining, focal loss, and mixup are integrated to further boost the performance. The contributions are 1. explore the fine-tuning problem for CSL. 2. provide the theoretical analysis 3. good experiment result.

**Ethical Concerns:**

No ethical concerns.

**Limitations And Societal Impact:**

1. Limited novelty. As explained in the Main Review.

2. Heavy hyperparameters tuning: The proposed approach involves many hyperparameters ($\eta$, $\alpha$, $\lambda_n$). Among them, $\eta$ and $\alpha$ are not consistent among datasets and requires grid search for EACH dataset. Besides, a different set of ($\eta$ and $\alpha$) may lead to different optimized learning rate, which makes the hyper-parameters even harder to tune. This greatly limits the practicability of this work.


**Main Review:**

Originality: The novelty of this work is limited. It is more like a simple extension of several existing techniques: there are existing works that have explored adding contrastive loss as a regularization in NLP [1]. Hard sample mining is also explored in many contrastive learning works [2][3], [2] also employs mix-up for generating hard samples.

Quality: The overall quality is good. There are many experiments supporting the claim.

Clarity: This work is well written. The descriptions are overall clear and easy to understand.

Significance: The significance of this work is limited: it involves heavy hyperparameters tuning, which limits its practicability. (Would explain in detail in the Limitations section).

Citation:
[1] Supervised Contrastive Learning for Pre-trained Language Model Fine-tuning; ICLR 2021
[2] Hard Negative Mixing for Contrastive Learning; Neurips 2020
[3] Contrastive learning with hard negative samples; ICLR 2021

**Time Spent Reviewing:**

4

---

> ### Author Response · Authors · 2021-08-10
> **Response**
>
> Thank you for your constructive review. We are glad to see that the overall quality, extensive experiments and good writing of this paper are appreciated. We address the main concerns point by point below.
>
> ***
> **Q1. The novelty of this work and its difference to existing studies [1,2,3].**
>
> A1. In the submission, we have discussed the differences between our proposed method and existing methods in Introduction (Lines 50-56 for the work [1]) and Related Work (Lines 88-94 for the work [2]). Here, we further highlight the differences.
>
> There are two key differences between our work and the work [1]:
>
> 1. Theoretical part: The work [1] only empirically explores contrastive learning in model fine-tuning but does not present any theoretical explanation on why it helps fine-tuning. In contrast, **our work theoretically proves** that optimizing the supervised contrastive loss benefits both discriminative representation learning and model optimization during fine-tuning in Lines 130-166.
>
> 2. Technical part: The work [1] only **simply adds the contrastive loss to the objective of fine-tuning,** without considering an important challenge in contrastive fine-tuning (i.e., hard pair mining) and thus cannot fine-tune contrastive self-supervised visual models well (cf. Table 1). In contrast, Core-tuning further develops a **new hard sample pair mining strategy** that is demonstrated to be more effective for contrastive fine-tuning. Besides, we also propose to **smooth the decision boundary** for better exploiting the learned discriminative feature space.
>
>
> Moreover, our work is also different from the studies [2,3]:
>
> 1. Task: The studies [2,3] focus on contrastive self-supervised **pre-training,** while our work explores the **fine-tuning** of contrastive self-supervised models.
>
> 2. Technical part: The two studies [2,3] seek to mine hard negative pairs for better self-supervised instance discrimination. Due to the lack of labels, they cannot accurately generate hard pairs regarding different classes. In contrast, by using data labels to differentiate positive and negative pairs, our proposed hardness-directed sample mixup strategy is able to **generate accurate hard positive and negative pairs for each class.** Note that accurate hard pair generation regarding classes is important to model fine-tuning for downstream tasks. In addition, our method further applies a **new focal contrastive loss** for better hard pair learning.
>
> In addition to the above differences, there are two more novelties in our work:
>
> 1. To our best knowledge, we are among the first to look into the fine-tuning stage of contrastive self-supervised models, which is an important yet under-explored problem. The task significance has been recognized by the reviewer XXej that **“This paper focuses on a real and practical problem in self-supervised learning”.**
>
> 2. We theoretically analyze the benefits of the supervised contrastive loss on representation learning and model optimization, revealing that it is beneficial to model fine-tuning. The significance has been recognized by the reviewer KA1Y that “**The theoretical bounds are both reasonable and give insights to the problem”.** We expect the theoretical result can motivate more work in the future.
>
>
> [1] Supervised Contrastive Learning for Pre-trained Language Model Fine-tuning. ICLR, 2021.
>
> [2] Hard Negative Mixing for Contrastive Learning. NeurIPS, 2020.
>
> [3] Contrastive learning with hard negative samples. ICLR, 2021.
>
> ***
> **Q2. The practicability of Core-tuning is limited by the tuning of hyper-parameter ($\eta$, $\alpha$, $\lambda_n$).**
>
> A2. Considering the hyper-parameters ($\eta$, $\alpha$, $\lambda_n$) in Core-tuning, only two of them (i.e., $\eta$ and $\alpha$) require tuning via cross-validation and their values are chosen from a small set of {0.1, 1, 10}, while $\lambda_n$ is fixed as 0.8.  Moreover, as shown in Figure 5 in Appendix E, Core-tuning is insensitive to the value choice of $\eta$ and $\alpha$. In fact, simply setting $\eta=0.1$ and $\alpha=1$ without careful tuning already gives good performance across all the datasets.
>
> Thanks for your kind consideration. We welcome and are very happy to discuss any further questions.

---

### Official Review · Reviewer_XXeJ · 2021-07-17

**Rating:** 6
**Confidence:** 4

**Summary:**

This paper focuses on how to fine-tune Contrastive Self-supervised Learning (CSL) models. In particular, it proposes a new fine-tuning method based on a hard-pair mining strategy. The authors provide a theoretical analysis to justify the benefits of supervised contrastive loss and then demonstrate empirically.


**Limitations And Societal Impact:**

The authors states that the potential limitation of the proposed methods is on the limited application range.

**Main Review:**

Strength:
1. This paper focuses on a real and practical problem in Self-supervised Learning.
2. This paper shows that contrastive loss has regularization effect on representational learning and can improve the optimization theoretically.
3. The paper is well-written and easy to follow, and the authors have done a good job of empirically analyzing the effect of proposed fine-tuning strategy in a number of different scenarios.

Weakness:
1. In section 4.1, the paper states that the generated negative pairs are semi-hard because the hard negatives may cause performance degradation, but there is little analysis about the choice given here, either empirical or theoretical.
2. Recent SSL work, e.g. BYOL[1], achieves SOTA performance without using any negative samples. I wonder whether the proposed method can be used for fine-tuning such SSL models.

[1]G, J.-B. et al. Bootstrap your own latent - a new approach to self-supervised learning, NeurIPS, 2020.



**Time Spent Reviewing:**

2

---

> ### Author Response · Authors · 2021-08-10
> **Response: More empirical results that further demonstrate the effectiveness of our method**
>
> Thanks very much for your valuable comments on our paper. We address the main concerns below.
> ***
> **Q1. Analysis about the choice of semi-hard negative pairs.**
>
> A1.  As mentioned in Lines 192-194, selecting the hardest negative sample pair to generate harder negative pairs may result in false negatives if there exist outliers, leading to bad local minima in training and performance degradation in practice. In contrast, selecting semi-hard negative pairs helps to overcome potential noise and thus yields better performance. In fact, the effectiveness of semi-hard negative pairs has been demonstrated in metric learning [1-3]. To verify this, we further evaluate the performance of Core-tuning with the strategy that generates the hardest negative pairs. The below table shows that Core-tuning with semi-hard negatives performs better than Core-tuning with hard negatives.
>
>
> |  Model   | Pre-training  | Fine-tuning  | Caltech101  | DTD  | Pets
> |  ----  | :----: | ---- | :----: | :----: | :----: |
> |  ResNet-50  |     MoCo-v2  |       ours (hard negatives)            |        92.75       |     74.79    |     91.36 |
> |  ResNet-50  |     MoCo-v2   |       ours (semi-hard negatives)  |         93.46     |       75.37   |      92.36 |
>
> &nbsp;
>
> [1] FaceNet: A Unified Embedding for Face Recognition and Clustering. CVPR, 2015.
>
> [2] Deep metric learning via lifted structured feature embedding. CVPR, 2016.
>
> [3] Sampling Matters in Deep Embedding Learning. CVPR, 2017.
>
> ***
>
> **Q2. More results on BYOL pre-trained models.**
>
> A2. Following the suggestion, we further conduct experiments on BYOL pre-trained models. As shown in the following table, Core-tuning also improves  BYOL pre-trained ResNet-50 to perform better on downstream tasks, compared to the conventional CE-tuning. We will add the results in the revision.
>
> |  Model   | Pre-training  | Fine-tuning  | Caltech101  | DTD  | Pets
> |  ----  | :----: | :----: | :----: | :----: | :----: |
> |  ResNet-50  |     BYOL  |       CE-tuning            |        91.19   |   	74.94  |  	92.39 |
> |  ResNet-50  |     BYOL   |       Core-tuning  |        93.25     | 	76.56   | 	93.74 |
>
> Thanks for your kind consideration. We are very happy to address any further questions you may have.

---

### Decision · Program_Chairs · 2021-09-27

**Decision:**

Accept (Poster)

**Comment:**

This paper investigates ways to improve the training of self-supervised models on downstream tasks by adding a contrastive loss to the fine-tuning process as well as augmenting this loss with methods for hard negative and hard positive pair mining. The authors show theoretically that a supervised contrastive loss benefits performance and show empirically that this method results in improved performance on a number of downstream datasets. Reviewers agreed that the clarity and quality of the work was high, though there were some concerns regarding the novelty of the proposed approach. While I agree with reviewers that this paper combines several previously known innovations rather than introducing a completely novel approach, I do not think this will limit the impact of the paper, as there is substantial value in combining previously independent observations. As such, I recommend the paper be accepted as a poster.